# ACE2 binding is an ancestral and evolvable trait of sarbecoviruses

Tyler N. Starr[1,2,6 ✉], Samantha K. Zepeda[3,6], Alexandra C. Walls[3], Allison J. Greaney[1,4], Sergey Alkhovsky[5], David Veesler[2,3 ✉] & Jesse D. Bloom[1,2,4 ✉]

Two different sarbecoviruses have caused major human outbreaks in the past two decades[1,2]. Both of these sarbecoviruses, SARS-CoV-1 and SARS-CoV-2, engage ACE2 through the spike receptor-binding domain[2–6]. However, binding to ACE2 orthologues of humans, bats and other species has been observed only sporadically among the broader diversity of bat sarbecoviruses[7–11]. Here we use high-throughput assays[12] to trace the evolutionary history of ACE2 binding across a diverse range of sarbecoviruses and ACE2 orthologues. We find that ACE2 binding is an ancestral trait of sarbecovirus receptor-binding domains that has subsequently been lost in some clades. Furthermore, we reveal that bat sarbecoviruses from outside Asia can bind to ACE2. Moreover, ACE2 binding is highly evolvable—for many sarbecovirus receptor-binding domains, there are single amino-acid mutations that enable binding to new ACE2 orthologues. However, the effects of individual mutations can differ considerably between viruses, as shown by the N501Y mutation, which enhances the human ACE2-binding affinity of several SARS-CoV-2 variants of concern[12] but substantially decreases it for SARS-CoV-1. Our results point to the deep ancestral origin and evolutionary plasticity of ACE2 binding, broadening the range of sarbecoviruses that should be considered to have spillover potential.

Both SARS-CoV-2 and SARS-CoV-1 use human ACE2 as their receptor[2–6]. Sampling of bats has identified multiple lineages of sarbecoviruses with receptor-binding domains (RBDs) exhibiting different ACE2-binding properties[7–11,13–19] that are exchanged through recombination[8,19,20]. Before the emergence of SARS-CoV-2, all bat sarbecoviruses with a demonstrated ability to bind to any ACE2 orthologue contained RBDs related to SARS-CoV-1 and were sampled from *Rhinolophus sinicus* and *Rhinolophus affinis* bats in Yunnan province in southwest China[7,8,11,21,22]. More recently, sarbecoviruses related to SARS-CoV-2 that bind to ACE2 have been found more widely across Asia and from a broader diversity of *Rhinolophus* species[2,16,23–25]. However, ACE2 binding has not been observed within a prevalent group of sarbecovirus RBDs sampled in southeast Asia (RBD clade 2)[7,8,17], nor has it been observed in distantly related sarbecoviruses from Africa and Europe (RBD clade 3)[7,19] (Fig. 1a). It is therefore unclear whether ACE2 binding is an ancestral trait of sarbecovirus RBDs that has been lost in some RBD lineages, or a trait that was acquired more recently in a subset of Asian sarbecovirus RBDs[19,20]. As ACE2 is also variable among *Rhinolophus* bats, particularly in the surface recognized by sarbecoviruses[26–28], it is important to understand how sarbecoviruses acquire the ability to bind to new ACE2 orthologues, including that of humans, through amino acid mutations.

### Survey of sarbecovirus ACE2 binding

To trace the evolutionary history of sarbecovirus binding to ACE2, we assembled a gene library encoding 45 sarbecovirus RBDs spanning

all four known RBD phylogenetic clades (Fig. 1a, b and Extended Data Fig. 1). We cloned the RBD library into a yeast-surface display platform that enables high-throughput measurement of ACE2-binding avidities using titration assays combining fluorescence-activated cell sorting (FACS) and deep sequencing[12] (Extended Data Fig. 2a–d). We also assembled a panel of recombinant, dimeric ACE2 proteins from human, civet, pangolin and mouse, as well as two alleles each from *R. affinis* and *R. sinicus* bats[26] (Fig. 1c). The *R. affinis* alleles encode the two distinct RBD-interface sequences found among 23 *R. affinis* bats from Yunnan and Hubei, China. The *R. sinicus* alleles encode two out of the eight distinct RBD-interface sequences found among 25 *R. sinicus* bats from Yunnan, Hubei, Guangdong and Guangxi provinces, and Hong Kong[26], including one allele (3364) that is closest to consensus among the 8 RBD-interface sequences, and another (1434) that does not support entry by some clade 1a sarbecoviruses[26]. We measured the apparent dissociation constant ($K_{D,app}$) of each RBD for each of the eight ACE2 orthologues (Fig. 1b, d and Extended Data Fig. 2). We performed all of the experiments in duplicate using independently constructed libraries, and the measurements were highly correlated between replicates ($R^2 > 0.99$; Extended Data Fig. 2g).

Consistent with a previous survey of human ACE2-mediated cellular infectivity[7], human ACE2 binding is restricted to RBDs within the SARS-CoV-1 and SARS-CoV-2 clades (Fig. 1b), although binding affinities vary among RBDs within these clades. Specifically, the RBDs from SARS-CoV-2 and related viruses from pangolins bind to human ACE2 with high affinity, whereas the RBD from the bat virus RaTG13

[1]Basic Sciences Division, Fred Hutchinson Cancer Research Center, Seattle, WA, USA. [2]Howard Hughes Medical Institute, Seattle, WA, USA. [3]Department of Biochemistry, University of Washington, Seattle, WA, USA. [4]Department of Genome Sciences, University of Washington, Seattle, WA, USA. [5]N.F. Gamleya National Center of Epidemiology and Microbiology, Ministry of Health of the Russian Federation, Moscow, Russia. [6]These authors contributed equally: Tyler N. Starr, Samantha K. Zepeda. ✉e-mail: tstarr@fredhutch.org; dveesler@uw.edu; jbloom@fredhutch.org

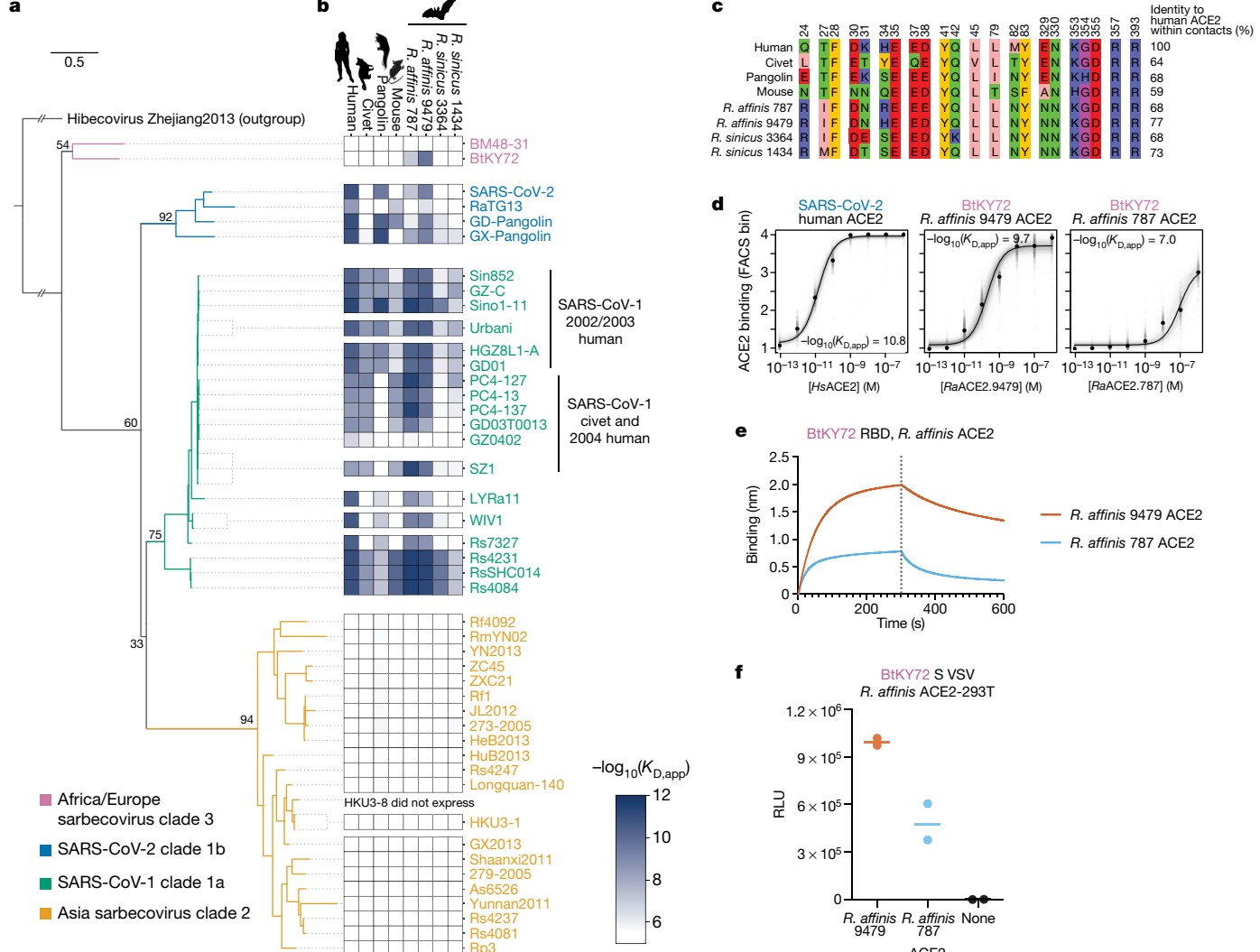

**Fig. 1 | High-throughput survey of sarbecovirus ACE2 binding. a**, Maximum likelihood phylogeny of sarbecovirus RBDs constructed from RBD nucleotide sequences. The node labels indicate bootstrap support values. Details on rooting are shown in Extended Data Fig. 1. Scale bar, 0.5 nucleotide substitutions per site. **b**, Binding avidities of sarbecovirus RBDs for eight ACE2 orthologues determined using high-throughput yeast-displayed RBD titration assays (Extended Data Fig. 2). **c**, Alignment of tested ACE2 orthologues within RBD-contact positions (4 Å cut-off in Protein Data Bank (PDB) 6M0J or 2AJF). **d**, Representative ACE2-binding curves from high-throughput titrations. Underlying titration curves for individual replicate-barcoded representatives of a genotype are shown in light grey, and the average binding across all barcodes is indicated in black.

**e**, BLI binding analysis of 1 μM *R. affinis* ACE2–Fc binding to biotinylated BtKY72 RBD immobilized at the surface of streptavidin biosensors (see Extended Data Fig. 3a for analysis of the robustness of the result to ACE2–Fc concentration). Data are representative of three assays using independent preparations of RBD (biological triplicate). **f**, Entry of VSV particles pseudotyped with the BtKY72 spike into HEK293T cells transiently expressing *R. affinis ACE2* alleles 9479 or 787. Each point represents the mean of technical triplicates for assays performed with independent preparation of pseudoviral particles (biological duplicate). The geometric mean is shown by the horizontal line. The normalized pseudovirus western blot, and mock (VSV prepared without spike plasmid) pseudovirus entry in *R. affinis* ACE2 HEK293T cells are shown in Extended Data Fig. 3c, d.

exhibits much lower affinity[16]. The RBDs of SARS-CoV-1 isolates from the 2002–2003 epidemic bind to human ACE2 strongly, whereas RBDs from civet and sporadic 2004 human isolates (GD03T0013, GZ0402) show weaker binding, consistent with their civet origin and limited transmission[29,30]. SARS-CoV-1-related bat virus RBDs bind to human ACE2, in some cases with higher affinity than SARS-CoV-1 itself.

Binding to civet ACE2 was detected only within the SARS-CoV-1 clade, whereas pangolin ACE2 binding is more widespread within the SARS-CoV-2 clade, consistent with viruses isolated from civet or pangolin partitioning specifically within each of these clades. Mice are not a natural host of sarbecoviruses, and RBDs from the SARS-CoV-1 and SARS-CoV-2 clades bind to mouse ACE2 only sporadically, typically with modest to weak affinity relative to other ACE2 orthologues. The highest binding affinity for mouse ACE2 is found in the cluster of RBDs related to RsSHC014, which can mediate infection and pathogenesis in mice[31].

Binding to ACE2 of *R. affinis* and particularly *R. sinicus* bats varies considerably among strains in the SARS-CoV-1 and SARS-CoV-2 clades, consistent with an evolutionary arms race driving ACE2 variation in *Rhinolophus* bats[26,27]. The two *R. sinicus* bat ACE2 proteins tested interacted only with SARS-CoV-1 isolates and the bat RsSHC014-cluster RBDs, which are notable for their broad ACE2-binding specificity in our assay. By contrast, we detected strong binding to both *R. affinis* ACE2 proteins among many RBDs in the SARS-CoV-1 and SARS-CoV-2 clades. However, the RBDs of the two viruses sampled from *R. affinis* in our panel bound only modestly (LYRa11) or very weakly (RaTG13) to the *R. affinis* ACE2s that we tested.

Strikingly, we detected binding to *R. affinis* ACE2 proteins by the RBD of the BtKY72 virus from Kenya[13] (Fig. 1b, d), the first described binding to any ACE2 orthologue for a sarbecovirus outside of Asia[7,19]. To validate this finding, we purified the BtKY72 RBD and *R. affinis* ACE2–Fc fusion

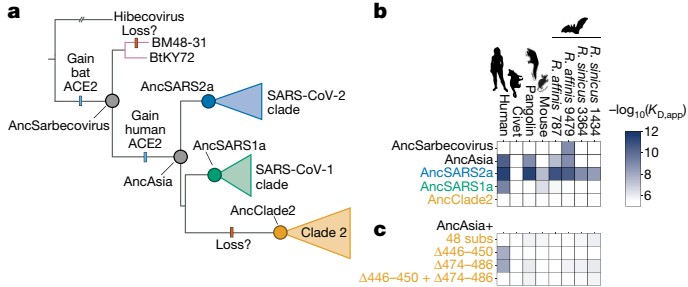

**Fig. 2 | Ancestral origins of sarbecovirus ACE2 binding. a**, Clade-collapsed RBD phylogeny. The circles represent nodes at which ancestral sequences were inferred. The bars indicate putative gains and losses in ACE2 binding. **b**, ACE2 binding of ancestrally reconstructed, yeast-displayed RBDs (Extended Data Figs. 5 and 6). **c**, ACE2 binding of AncAsia RBD plus introduction of the 48 substitutions or 2 sequence deletions that occurred on the phylogenetic branch leading to AncClade2 RBD.

proteins recombinantly expressed in human cells and characterized their interaction using biolayer interferometry (BLI). In agreement with the yeast-display results, the BtKY72 RBD bound to the *R. affinis* 9479 ACE2 and more weakly to the *R. affinis* 787 allele (Fig. 1e and Extended Data Fig. 3a). Furthermore, HEK293T cells transfected with the *R. affinis* 9479 or 787 *ACE2* alleles supported the entry of vesicular stomatitis virus (VSV) particles pseudotyped with the BtKY72 spike, thereby demonstrating that ACE2 is a bona fide entry receptor for this virus (Fig. 1f and Extended Data Fig. 3c, d). The geographical range of *R. affinis* does not extend outside of Asia[15], but this result indicates that BtKY72 may bind to ACE2 orthologues of bats found in Africa, although the full range of non-Asian bat species that harbour sarbecoviruses and their ACE2 sequences are underexplored[13,14,19,32].

We did not detect ACE2 binding by any of the clade 2 RBDs. In our panel, 9 out of the 23 clade 2 RBDs were sampled from *R. sinicus*, in some cases from the same caves—and even found co-infecting the same *R. sinicus* bats[8]—as ACE2-utilizing SARS-CoV-1-related RBDs. We tested binding by two clade 2 RBDs isolated from *R. sinicus* (YN2013 from Yunnan and HKU3-1 from Hong Kong Special Administrative Region) to an expanded ACE2 panel comprising all RBD-interface sequences observed in *R. sinicus* bats[26], including those sampled in Yunnan and Hong Kong Special Administrative Region. In contrast to SARS-CoV-1 Urbani and RsSHC014 (a clade 1a RBD isolated from *R. sinicus* in Yunnan[11]), YN2013 and HKU3-1 RBDs did not bind to any of the eight *R. sinicus* ACE2 proteins (Extended Data Fig. 4). Previous experiments with clade 2 RBDs have also demonstrated a lack of binding to *R. pearsonii*[17] and human[7,8,12,17] ACE2. Clade 2 RBDs have two large deletions within the receptor-binding motif[7,8,19], which has led to the hypothesis that this clade uses an unidentified alternative receptor, which could be bound by either the RBD or the spike N-terminal domain[33–36]. Our results are consistent with this hypothesis, although we cannot rule out that clade 2 RBDs bind to other ACE2 orthologues that have not yet been tested.

## Ancestral origins of ACE2 binding

Our finding that the BtKY72 RBD binds to ACE2 suggests that ACE2 binding was present in the ancestor of all sarbecoviruses before the split of Asian and non-Asian RBD clades (Fig. 2a). To test this hypothesis, we used ancestral sequence reconstruction[37] to infer plausible sequences representing ancestral nodes on the sarbecovirus RBD phylogeny (Fig. 2a and Extended Data Fig. 5a). We evaluated ACE2 binding for the most probable reconstructed ancestral sequences (Fig. 2b and Extended Data Fig. 5b) and in alternative reconstructions that incorporate statistical or phylogenetic ambiguities inherent to ancestral reconstruction (Extended Data Fig. 6). Consistent with the distribution of ACE2 binding among extant sarbecoviruses, the reconstructed

ancestor of all sarbecovirus RBDs (AncSarbecovirus) bound to the *R. affinis* 9479 ACE2 (Fig. 2b). Broader ACE2 binding (including to human ACE2) was acquired on the branch connecting AncSarbecovirus to the ancestor of the three Asian sarbecoviruses RBD clades (AncAsia). ACE2 binding was then lost along the branch to the clade 2 ancestor (Anc-Clade2), due to the combination of 48 amino-acid substitutions and 2 deletions within the ACE2-binding region that occurred along this branch (Fig. 2c).

This evolutionary history of ACE2 binding is robust to some but not all analyses of uncertainty in our phylogenetic reconstructions[38,39]. The key phenotypes represented in Fig. 2b are robust to uncertainties in the topology of the RBD phylogeny (Extended Data Fig. 6a, b) or possible recombination within the RBD impacting the cluster of RBDs related to RsSHC014 (Extended Data Fig. 6c–f). However, statistical uncertainty in the identity of some ACE2-contact positions affects our inferences, with some reasonably plausible 'second-best' reconstructed states altering ancestral phenotypes (Extended Data Fig. 6b). Nonetheless, our hypothesis of an ancestral origin of sarbecovirus ACE2 binding is supported by the most plausible ancestral reconstructions as well as the distribution of ACE2 binding among the directly sampled sarbecovirus RBDs in Fig. 1a, b.

## Evolvability of ACE2 binding

To examine how easily RBDs can acquire ACE2 binding through single amino-acid mutations, we constructed mutant libraries in 14 RBD backgrounds spanning the RBD phylogeny. In each background, we introduced all single amino acid mutations at six RBD positions previously implicated in the evolution of receptor binding in SARS-CoV-2 and SARS-CoV-1 (refs. [12,40]) (SARS-CoV-2 residues Leu455, Phe486, Gln493, Ser494, Gln498 and Asn501; Fig. 3a; we use SARS-CoV-2 numbering for mutations in all of the homologues below). We recovered nearly all 1,596 of the intended mutations, and measured the binding of each mutant RBD to each ACE2 orthologue using high-throughput titrations as described above.

The results show that ACE2 binding is a remarkably evolvable trait (Fig. 3b, c and Extended Data Fig. 7). In almost all cases in which a parental RBD binds to a particular ACE2, there are single amino acid mutations that improve binding by greater than fivefold. Thus, ACE2 binding can easily be enhanced by mutation, which may facilitate the frequent host jumps seen among sarbecoviruses[41]. Notably, our data on mouse ACE2 binding could inform the development of mouse-adapted sarbecovirus strains for in vivo studies[31,42–44], including potentially safer strains that bind to mouse but not human ACE2 (Extended Data Fig. 8).

In the majority of cases in which an RBD does not bind to a particular ACE2 orthologue, single mutations can confer low to moderate binding affinity (Fig. 3b, c). The only exceptions are BM48-31 and AncClade2, for which none of the tested mutations enabled binding to any of the ACE2 variants. We found that the mutation K493Y in AncSarbecovirus enables binding to human ACE2 (Fig. 3b and Extended Data Fig. 7), although this particular mutation did not occur on the branch to AncAsia where we inferred that human ACE2 binding was historically acquired, illustrating the existence of multiple evolutionary paths to acquiring human ACE2 binding. We identified single mutations at positions 493, 498 and 501 that enable the BtKY72 RBD to bind to human ACE2 (Fig. 3b and Extended Data Fig. 7), suggesting that human ACE2 binding is evolutionarily accessible in this lineage.

We validated that the mutations K493Y and T498W enable the RBD of the African sarbecovirus BtKY72 to interact with human ACE2 using purified recombinant proteins. Binding to human ACE2–Fc is not detectable with the parental BtKY72 RBD using BLI but is conferred by T498W and enhanced for the K493Y/T498W double mutant (Fig. 3d and Extended Data Fig. 3b). To evaluate whether the observed binding translated into cell entry, we generated VSV particles pseudotyped with the wild-type or mutant BtKY72 spikes and tested entry in HEK293T cells expressing human ACE2. We detected robust spike-mediated

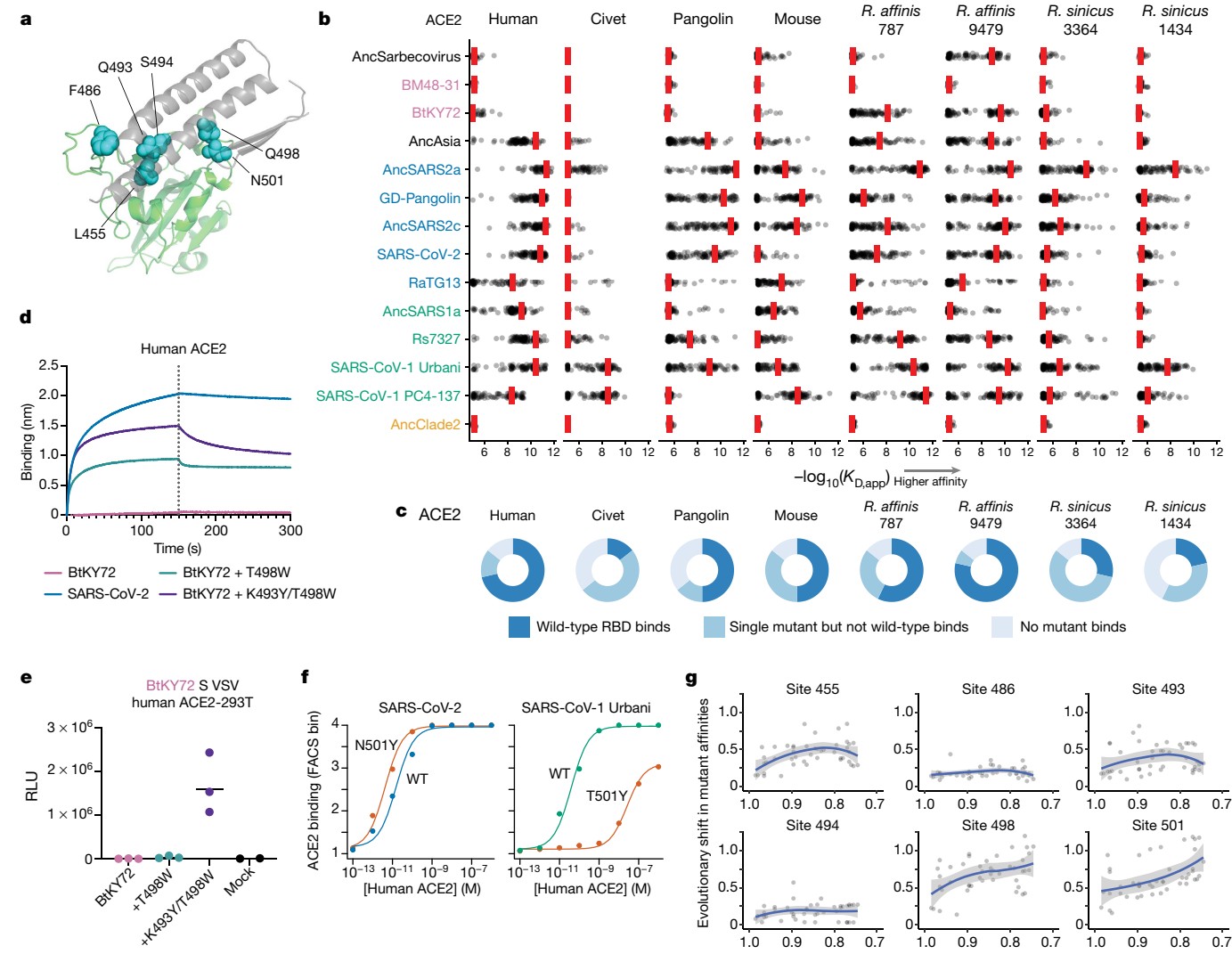

**Fig. 3 | Evolutionary plasticity of ACE2 binding. a**, The structural context of positions targeted for mutagenesis. Green cartoon, RBD; grey cartoon, ACE2 interaction motifs; blue spheres, residues targeted through mutagenesis (SARS-CoV-2 identities). **b**, Mutational scanning measurements. The red bars mark the binding avidity of the parental RBD, and the points mark mutant avidities (see Extended Data Fig. 7 for mutation-level measurements). **c**, The fraction of the 14 RBD backgrounds for which the parental RBD binds to the indicated ACE2 orthologue ($-\log_{10}(K_{D,app}) > 7$), a single mutant binds but the parental RBD does not, or no tested mutants bind. **d**, Binding of 1 μM human ACE2–Fc to biotinylated RBDs immobilized at the surface of streptavidin biosensors (see Extended Data Fig. 3b for an analysis of the robustness of the result to ACE2–Fc concentration). Data are representative of three assays using independent preparations of RBD (biological triplicate). **e**, Entry of BtKY72 spike-pseudotyped VSV in HEK293T cells stably expressing human ACE2. Each point represents the mean of technical triplicates in assays performed with independent preparation of pseudoviral particles (biological triplicate). The horizontal line shows the geometric mean. Mock, VSV particles produced in cells in which no spike gene was transfected. A western blot of pseudotyped particles is shown in Extended Data Fig. 3c, and entry into HEK293T cells lacking ACE2 is shown in Extended Data Fig. 3e. **f**, Titration curves illustrating the effect of mutation to tyrosine 501 (SARS-CoV-2 numbering) in the SARS-CoV-2 and SARS-CoV-1 Urbani RBD backgrounds. **g**, Epistatic turnover in mutation effects. Each point represents, for a pair of RBDs, the mean absolute error (residual) in their correlated mutant avidities for human ACE2 (Extended Data Fig. 9a) versus their pairwise amino acid sequence identity. Correlations were computed only for pairs in which the parental RBDs bind with $-\log_{10}(K_{D,app}) > 7$. Data are LOESS mean (blue line) ± 95% confidence intervals trendline (grey shading) (see Extended Data Fig. 9b for an analysis across all ACE2 orthologues).

entry for the K493Y/T498W double mutant but not the T498W single mutant (Fig. 3e and Extended Data Fig. 3c, e), reflecting their apparent avidities (Fig. 3d) and confirming the evolvability of human ACE2 binding in this African sarbecovirus lineage.

Finally, we examined how the mutations that enhance ACE2 binding differ among sarbecovirus backgrounds, reflecting epistatic turnover in mutation effects[12,45]. For example, the N501Y mutation increases human ACE2-binding affinity for SARS-CoV-2 where it has arisen in variants of concern[46], but the homologous mutation in the SARS-CoV-1 RBD (position 487) is highly deleterious for human ACE2 binding (Fig. 3f). More broadly, variation in mutant effects increases as RBD sequences diverge (Fig. 3g

and Extended Data Fig. 9). However, the rate of this epistatic turnover varies across positions—for example, the effects on human ACE2 binding for mutations at positions 486 and 494 remain relatively constant across sequence backgrounds, whereas variability in the effects of mutations at positions 498 and 501 increases substantially as RBDs diverge.

## New sarbecovirus lineages bind to ACE2

Given that ACE2 binding is an ancestral sarbecovirus trait with plastic evolutionary potential, unsampled sarbecoviruses lineages probably have the ability to bind to ACE2 and evolve to bind to human ACE2 unless

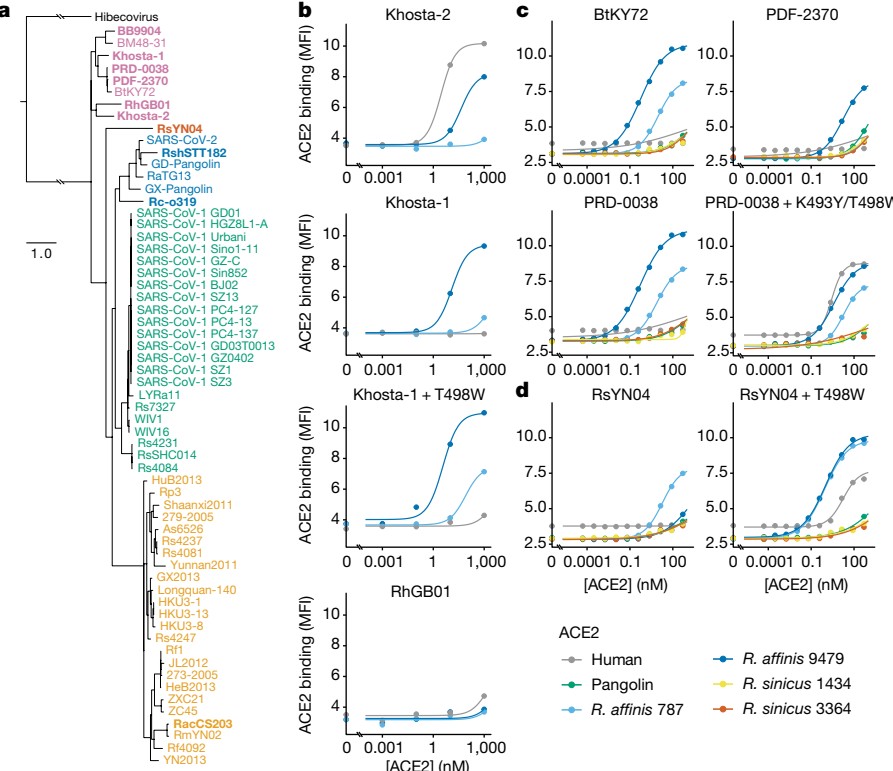

**Fig. 4 | Newly sampled sarbecovirus lineages bind to ACE2. a,** Phylogenetic placement of the newly described sarbecovirus RBDs. The new sequences are shown in bold font. RBDs are coloured according to the key in Fig. 1a (Extended Data Fig. 10). Scale bar, expected nucleotide substitutions per site.
**b–d,** Binding curves for newly described sarbecovirus RBDs from Europe (**b**), Africa (**c**) and Asia (**d**), and candidate mutations that confer human ACE2 binding. Measurements were performed with yeast-displayed RBDs and purified dimeric ACE2 proteins, measured using flow cytometry. Data are from a single experimental replicate.

these traits have been specifically lost as occurred in clade 2. To test this idea, we investigated sarbecoviruses reported after the initiation of our study, including viruses from Africa[19] and Europe[32,47] and a new RBD lineage represented by RsYN04 from a *Rhinolophus stheno* bat in Yunnan, China[15], which branches separately from the four RBD clades previously described (Fig. 4a).

We determined the ACE2-binding abilities of these RBDs using our yeast-display platform. We found that two newly described sarbecoviruses from the Caucasus region of Russia[32] bind to ACE2 (Fig. 4b): the Khosta-1 RBD binds to *R. affinis* ACE2s with avidity that is improved by the T498W mutation and, strikingly, the Khosta-2 RBD binds to human ACE2 even in the absence of mutations. The Khosta-2 RBD was also recently shown to enable cell entry through human ACE2 (refs. [48,49]). This finding indicates that the evolvability of human ACE2 binding that we describe for other African and European sarbecoviruses has been realized in naturally circulating viruses that are geographically and phylogenetically separated from the southeast Asian clades from which spillover has been described to date. Our results also reinforce our observation of ACE2 binding in African sarbecoviruses (Fig. 4c)— similar to BtKY72, RBDs of the newly described African sarbecoviruses PDF-2380 and PRD-0038 (ref. [19]) bind to *R. affinis* ACE2s, and the K493Y/T498W double mutant confers human ACE2 binding to the PRD-0038 RBD as it does for BtKY72. Finally, the uniquely branching RsYN04 RBD binds to *R. affinis* 787 ACE2 (Fig. 4d), as was recently shown for the closely related RaTG15 spike[50]. The RsYN04 RBD can also acquire binding to human ACE2 through the single T498W mutation. Incorporation of newly described sarbecovirus sequences into an updated phylogenetic reconstruction of the AncSarbecovirus RBD sequence reaffirms the conclusion that the ancestral sarbecovirus binds to bat ACE2 and can evolve human ACE2 binding through single

amino-acid mutation (Extended Data Fig. 10). These results illustrate that the ancestral traits of ACE2 binding and ability to evolve human ACE2 binding are maintained in geographically and phylogenetically diverse sarbecoviruses, including lineages that are just beginning to be described[13,15,19,32,50].

## Discussion

Our experiments reveal that binding to bat ACE2 is an ancestral trait of sarbecoviruses that is also present in viruses from outside of Asia[13,19,32]. Binding to human ACE2 arose in the common ancestor of SARS-CoV-1- and SARS-CoV-2-related RBDs before their divergence, and human ACE2 binding is evolvable in other phylogenetic clades. Binding to the ACE2 orthologues that we tested was then lost on the branch leading to the clade 2 RBDs, which either bind to an alternative receptor or ACE2 orthologues that were not evaluated here. These results imply that unsampled RBD lineages in the phylogenetic interval between BtKY72 and SARS-CoV-1/SARS-CoV-2 probably use ACE2 as an entry receptor and have the ability to evolve affinity for human ACE2. Indeed, the Khosta-2 virus from Russia provides an example of a RBD for which this evolutionary potential for human ACE2 binding has been realized.

Our research also shows that ACE2 binding is a highly evolvable trait of sarbecovirus RBDs. For every ACE2-binding RBD that we studied, there were single amino acid mutations that enhanced affinity for ACE2 orthologues that a RBD could already bind to or that conferred binding to new ACE2 orthologues from different species. Host jumps are common among the wide diversity of bats that are naturally infected with these viruses[8,15,41]. In addition to frequent exchange of RBDs among viral backbones through recombination[8,19,20], the evolutionary plasticity of RBD binding to ACE2 is probably a key contributor to the ecological

dynamics of sarbecoviruses, and perhaps other coronaviruses that frequently transmit across species[51]. As the effects of RBD mutations on ACE2 binding can differ across sarbecovirus backgrounds, it is not trivial to predict the ACE2-binding properties of a given RBD solely from its sequence. Thus, high-throughput approaches such as the one we have used here, which enables rapid and comprehensive measurement of ACE2-binding affinities of RBD variants in a non-viral context, can aid efforts to understand the evolutionary diversity and dynamics of sarbecoviruses and develop broadly protective therapeutics.

Sarbecoviruses are of particular concern, as two different strains have caused human outbreaks. Although human infectivity depends on many factors, the ability to bind to human receptors is certainly a key factor. Our results show that the ability of sarbecoviruses to bind to human ACE2 is evolvable and has arisen independently in regions outside of southeast Asia. Our high-throughput yeast-display platform enables the study of possible host tropism of sarbecoviruses without requiring work with replication-competent viruses that can pose biosafety concerns. The geographical breadth of ACE2 binding that we describe suggests that care should be taken in the sampling and study of replication-competent sarbecoviruses even outside regions such as southeast Asia in which spillover potential is considered greatest, and that efforts to develop vaccines and antibody therapeutics for pandemic preparedness should consider sarbecoviruses circulating worldwide.

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

## Methods

### Phylogenetics and ancestral sequence reconstruction

All steps of the bioinformatic analysis, including specific programmatic commands, alignments, raw data and output files are provided at GitHub (https://github.com/jbloomlab/SARSr-CoV_homolog_survey/tree/master/RBD_ASR).

A panel of unique sarbecovirus RBD sequences was assembled incorporating the RBD sequences curated in ref. [7], all unique RBD sequences among SARS-CoV-1 human and civet strains reported in ref. [30], and recently reported sarbecoviruses BtKY72 (ref. [13]), RaTG13 (ref. [2]) GD-Pangolin-CoV (consensus RBD sequence reported in figure 3a of ref. [23]) and GX-Pangolin-CoV[23] (P2V, ambiguous nucleotide in codon 515 (SARS-CoV-2 numbering) was resolved to retain amino acid Phe515, which is conserved across all other sarbecoviruses). We also incorporated newly described sarbecovirus sequences RsYN04 (ref. [15]), PDF-2370 and PRD-0038 (ref. [19]), Khosta-1 and Khosta-2 (ref. [32]), RhGB01 (ref. [47]), RshSTT182 (ref. [25]) and Rc-o319 (ref. [24]) into updated phylogenies and functional work after the initiation of our study (Fig. 4 and Extended Data Fig. 10). The Hibecovirus sequence Hp-BetaCoV/Zhejiang2013 (GenBank: KF636752) was used to root the sarbecovirus phylogeny. For Extended Data Figs. 1 and 10a–d, additional betacoronavirus outgroups were included in rooting. All virus names, species and location of sampling, and sequence accessions or citations are provided at GitHub (https://github.com/jbloomlab/SARSr-CoV_homolog_survey/blob/master/RBD_ASR/RBD_accessions.csv).

Amino acid sequences were aligned by mafft (v.7.471)[52] with a gap opening penalty of 4.5. RBD sequences were subsetted from spike alignments according to our domain boundary defined for SARS-CoV-2 (Wuhan-Hu-1 GenBank: MN908947, residues Asn331–Thr531). Nucleotide alignments were constructed from amino acid alignments using PAL2NAL (v.14)[53]. Phylogenies were inferred with RAxML (v.8.2.12)[54] using the LG+Γ substitution model for amino acid sequence alignments or GTR+Γ with separate data partitions applied to the first, second and third codon positions for nucleotide sequence alignments. Constraint files specifying specific clade relationships (but free topologies within clades) were used to fix particular topologies in Extended Data Fig. 6a (alternative relationships between RBD clades 1a, 1b and 2) and Fig. 4a (monophyletic Europe and Africa RBD clade; Extended Data Fig. 10a–d). RBD gene segments were used as our primary boundary for phylogenetic inference and ancestral sequence reconstruction due to the presence of frequent recombination within broader spike alignments[19,20].

Marginal likelihood ancestral sequence reconstruction was performed with FastML (v.3.11)[55] using the amino acid sequence alignment, the maximum likelihood nucleotide tree topology from RAxML, the LG+Γ substitution matrix, re-optimization of branch lengths and FastML's likelihood-based indel reconstruction model. The maximum a posteriori ancestral sequences at nodes of interest were determined from the marginal reconstructions as the string of amino acids at each alignment site with the highest posterior probability, censored by deletions as inferred from the indel reconstruction. To test the robustness of ancestral phenotypes to statistical uncertainty in reconstructed ancestral states, we also constructed 'alt' ancestors in which all second-most-probable states with posterior probability > 0.2 were introduced simultaneously[38].

To identify potential recombination breakpoints within the RBD alignment, we used GARD (v.0.2)[56], which identified a possible recombination breakpoint (Extended Data Fig. 6c) that produces two alignment segments exhibiting phylogenetic incongruence with a gain in overall likelihood sufficient to justify the duplication of phylogenetic parameters (ΔAIC = −85). To determine the impact of this possible recombination on ancestral sequence reconstructions, the alignment was split into separate segments at the proposed breakpoint. Phylogenies were inferred and ancestral sequences reconstructed on separate segments as described above, and reconstructed ancestral sequences at matched nodes for each segment were concatenated, as shown in Extended Data Fig. 6e.

### RBD library construction

Genes encoding all 73 unique extant and ancestral RBD amino acid sequences were ordered from Twist Bioscience, Genscript, and IDT. Gene sequences are provided at GitHub (https://github.com/jbloomlab/SARSr-CoV_homolog_survey/blob/master/RBD_ASR/parsed_sequences/RBD_sequence_set_annotated.csv). Genes were cloned in bulk into the pETcon yeast surface-display vector (plasmid 2649) as described previously[12]. As described in this previous publication, randomized N16 barcodes were appended by PCR downstream from RBD coding sequences. RBD sequences were pooled and barcoded in two independently processed replicates. The pooled, barcoded parental RBD libraries were electroporated into *Escherichia coli* and plated at an estimated bottleneck of ~22,000 colony-forming units, yielding an estimated ~300 barcodes per parental RBD within each library replicate.

In parallel, we cloned site saturation mutagenesis libraries of six positions in select RBD backgrounds. The positions targeted correspond to SARS-CoV-2 positions 455, 486, 493, 494, 498 and 501. The RBD-indexed position targeted in each background is provided at GitHub (https://github.com/jbloomlab/SARSr-CoV_homolog_survey/blob/master/RBD_ASR/parsed_sequences/RBD_sequence_set_annotated.csv). Precise site saturation mutagenesis pools were produced by Genscript, provided as plasmid libraries. Failed positions in the Genscript mutagenesis libraries (all six positions in SARS-CoV-1 Urbani, position 494 in SARS-CoV-2, and position 455 in RaTG13 and GD-Pangolin) or backgrounds chosen for mutagenesis subsequent to initial library design (BtKY72) were produced in-house by PCR-based mutagenesis using NNS degenerate mutagenic primers followed by Gibson Assembly of the mutagenized fragments. In duplicate, mutant libraries were pooled and N16 barcodes were appended downstream from the RBD coding sequence. The pooled, barcoded mutant libraries were electroporated into *E. coli* and plated at a target bottleneck corresponding to an average of 20 barcodes per mutant within each library replicate.

Colonies from bottlenecked transformation plates were scraped and plasmids were purified. Parental RBD and mutant pools were combined at ratios corresponding to expected barcode diversity, yielding the two separately barcoded library replicates used in high-throughput experiments. Plasmid libraries were transformed into yeast (AWY101 strain[57]) according to a previously described protocol[58], transforming 10 μg of plasmid at 10× scale.

### PacBio sequencing and analysis

As described previously[12], PacBio sequencing was used to acquire long sequence reads spanning the N16 barcode and RBD coding sequence. PacBio sequencing constructs were prepared from library plasmid pools by NotI digestion and gel purification, followed by SMRTbell ligation. Each library was sequenced across three SMRT Cells on a PacBio Sequel using 20 h video collection times. PacBio circular consensus sequences (CCSs) were generated from subreads using the ccs program (v.5.0.0), requiring 99.9% accuracy and a minimum of 3 passes. The resulting CCSs are available on the NCBI Sequence Read Archive (SRA), BioSample SAMN18316101.

CCSs were processed using alignparse (v.0.1.6)[59] to identify the RBD target sequence, call any mutations and determine the associated N16 barcode sequence, requiring no more than 18 nucleotide mutations from the intended target sequence, an expected 16-nucleotide-length barcode sequence and no more than 3 mismatches across the sequenced portions of the vector backbone.

We next used processed CCSs to link each barcode to the associated RBD sequence. We first filtered sequences with ccs-determined accuracies of <99.99% or indels. The empirical sequencing accuracy estimated by comparing RBD variants associated with barcode sequences sampled across multiple CCSs (https://jbloomlab.github.io/alignparse/

alignparse.consensus.html#alignparse.consensus.empirical_accuracy) was 99.0% and 98.4% in libraries 1 and 2, respectively. For barcodes sampled across multiple CCSs, we derived consensus RBD variant sequences, discarding barcodes of which CCSs with identical barcodes exhibited >1 point mutation or >2 indels, or of which >10% or >25% of CCSs with an identical barcode contained a secondary non-consensus mutation or indel, respectively. The CCS processing pipeline is available at GitHub (https://github.com/jbloomlab/SARSr-CoV_homolog_survey/blob/master/results/summary/process_ccs.md). The final barcode-variant lookup table, which links each N16 barcode with its associated RBD sequence, is available at GitHub (https://github.com/jbloomlab/SARSr-CoV_homolog_survey/blob/master/results/variants/nucleotide_variant_table.csv).

### ACE2 proteins for yeast-display assays
Recombinant dimeric ACE2 proteins for yeast-display binding assays were purchased or produced from commercial sources. Recombinant human ACE2 (UniProt: Q9BYF1-1) was purchased from ACROBiosystems (AC2-H82E6), consisting of residues 18–740 spanning an intrinsic dimerization domain, followed by a His tag and biotinylated Avitag used for downstream detection. Civet (*Paguma larvata*) ACE2 (UniProt: Q56NL1-1) was purchased from ACROBiosystems (AC2-P5248), consisting of residues 18–740 spanning an intrinsic dimerization domain, with an N-terminal His tag used for downstream detection. Mouse (*Mus musculus*) ACE2 (UniProt: Q8R0I0-1) was purchased from Sino Biological (50249-M03H), consisting of residues 18–740 spanning an intrinsic dimerization domain, followed by a His tag and human IgG1 Fc domain used for downstream detection.

The remaining ACE2s for yeast-display binding assays (with the exception of Extended Data Fig. 4) were produced by Genscript. Specifically, pangolin (*Manis javanica*, GenBank: XP_017505746.1), *R. affinis* 787 (GenBank: QMQ39222), *R. affinis* 9479 (GenBank: QMQ39227), *R. sinicus* 3364 (GenBank: QMQ39219) and *R. sinicus* 1434 (GenBank: QMQ39216) ACE2 residues 19–615 were cloned with a C-terminal human IgG1 Fc domain for dimerization and downstream detection. pcDNA3.4 expression plasmids were transfected into HD 293F cells for protein expression. ACE2–Fc fusion proteins were purified from day six culture supernatants by Fc-tag affinity purification.

### Library measurements of RBD expression and RBD⁺ enrichment
Transformed yeast library aliquots were grown overnight in a shaker at 30 °C in SD-CAA medium (6.7 g l$^{-1}$ yeast nitrogen base, 5.0 g l$^{-1}$ casamino acids, 2.13 g l$^{-1}$ MES and 2% (w/v) dextrose, pH 5.3). To induce RBD expression, yeast was washed and resuspended in SG-CAA + 0.1% D medium (6.7 g l$^{-1}$ yeast nitrogen base, 5.0 g l$^{-1}$ casamino acids, 2.13 g l$^{-1}$ MES, 2% (w/v) galactose and 0.1% (w/v) dextrose, pH 5.3) at an initial optical density at 600 nm ($OD_{600}$) of 0.67, and incubated at room temperature for 16–18 h with mild agitation.

For each library, 45 $OD_{600}$ of induced culture was washed twice with PBS-BSA (0.2 mg ml$^{-1}$), and RBD surface expression was labelled by a C-terminal c-Myc tag with 1:100 diluted FITC-conjugated chicken anti-c-Myc antibodies (Immunology Consultants Lab, CMYC-45F) in 3 ml PBS-BSA. Labelled cells were washed twice in PBS-BSA, and resuspended in PBS for FACS analysis.

Yeast library sorting experiments were conducted on the BD FAC-SAria II system with FACSDiva software (v.8.0.2). For high-throughput measurements of RBD expression levels, cells were gated for single cells (Extended Data Fig. 2b) and partitioned into 4 bins of FITC fluorescence (Extended Data Fig. 2c), where bin 1 captures 99% of unstained cells, and bins 2–4 split the remaining library population into tertiles. Cells were sorted into 5 ml tubes pre-wet with 1 ml of SD-CAA with 1% BSA. We recovered ~8 million cells per library across the 4 bins. Sorted cells were resuspended to $2 \times 10^6$ cells per ml in fresh SD-CAA with 1:100 penicillin–streptomycin, and grown overnight at 30 °C. Plasmids were purified from post-sort yeast samples of <$4 \times 10^7$ cells per miniprep

column using the Zymo Yeast Miniprep II kit (D2004) according to the manufacturer's instructions, with the addition of an extended (>2 h) Zymolyase treatment and a −80 °C freeze−thaw cycle before cell lysis. N16 barcodes were PCR amplified from each plasmid aliquot as described previously[12] and submitted for Illumina HiSeq 50 bp single-end sequencing.

To enrich properly expressing RBD variants for downstream titration experiments, we also sorted around $2 \times 10^7$ cells per library using the RBD⁺ (FITC⁺) bin (Extended Data Fig. 2b). RBD⁺-enriched populations were resuspended to $1 \times 10^6$ cells per ml for overnight outgrowth, and frozen at −80 °C in 9 $OD_{600}$ aliquots for subsequent titration experiments.

A pool of mutants that were added after the first set of experiments (mutations at position 455 in RaTG13 and GD-Pangolin, and mutations at all six positions in BtKY72) were not RBD⁺ enriched and were not part of the bulk expression Sort-seq measurement, but were pooled with the RBD⁺-enriched population of the primary libraries for subsequent titration assays.

### Library measurements of ACE2-binding affinities
For high-throughput measurements of ACE2-binding affinities, yeast libraries were induced for RBD expression as described above. Induced cultures were aliquoted at 8 $OD_{600}$ per titration sample and washed twice with PBS-BSA. Cells were resuspended across a range of ACE2 concentrations from $1 \times 10^{-6}$ M to $1 \times 10^{-13}$ M in 1 M intervals, plus a 0 M ACE2 concentration. The samples were incubated overnight at room temperature with mild agitation. The samples were washed twice in ice-cold PBS-BSA, and resuspended in 1 ml secondary label (1:100 Myc-FITC and 1:200 PE-conjugated streptavidin (Thermo Fisher Scientific, S866) for human ACE2, 1:200 iFluor647-conjugated mouse anti-His (Genscript, A01802) for civet ACE2 and 1:200 PE-conjugated goat anti-human IgG (Jackson ImmunoResearch Labs 109-115-098) for all other Fc-tagged ACE2 ligands), and incubated for 1 h on ice. Cells were washed twice with PBS-BSA and resuspended in PBS for FACS analysis.

Titration samples were binned for single RBD-expressing cells (Extended Data Fig. 2b), which were then partitioned into four bins on the basis of ACE2 binding (Extended Data Fig. 2d). At each concentration, a minimum of $5 \times 10^6$ cells were collected across the 4 bins. Sorted cells were resuspended in 1 ml SD-CAA with 1:100 penicillin–streptomycin, and grown overnight at 30 °C in deep-well plates. Plasmid aliquots from each population were purified using the Zymo Yeast 96-Well Miniprep kit (D2005) according to the manufacturer's instructions, with the addition of an extended (>2 h) Zymolyase treatment and a −80 °C freeze−thaw cycle before cell lysis. N16 barcodes were PCR amplified from each plasmid aliquot as described previously[12] and submitted for Illumina HiSeq 50 bp single-end sequencing.

For the pool of mutants that were added after the first set of experiments (mutations at position 455 in RaTG13 and GD-Pangolin, and mutations at all six positions in BtKY72), duplicate titrations were already conducted with the primary pool for human ACE2 and *R. affinis* 787 ACE2. Titrations with this smaller library sub-pool with these ACE2 ligands were conducted as described above, but scaled to 1.6 $OD_{600}$ per sample, collecting >1 million cells per concentration.

### Illumina barcode sequencing analysis
Demultiplexed sequence reads (available on the NCBI SRA, BioSample SAMN20174027) were aligned to library barcodes as determined from PacBio sequencing using dms_variants (v.0.8.5), yielding a count of the number of times each barcode was sequenced within each FACS bin. Read counts within each FACS bin were downweighted by the ratio of total reads from a bin compared to the number of cells that were actually sorted into that bin. The table giving downweighted counts of each barcode in each FACS bin is available at GitHub (https://github.com/jbloomlab/SARSr-CoV_homolog_survey/blob/master/results/counts/variant_counts.csv).

We estimated the RBD expression level of each barcoded variant on the basis of its distribution of counts across FACS bins and the known log-transformed fluorescence boundaries of each sort bin using a maximum likelihood approach[12,60], implemented with the fitdistrplus package (v.1.0.14)[61] in R. Expression measurements were retained for barcodes for which greater than 20 counts were observed across the four FACS bins. The full pipeline for computing per-barcode expression values is described at GitHub (https://github.com/jbloomlab/SARSr-CoV_homolog_survey/blob/master/results/summary/compute_expression_meanF.md).

We estimated the level of ACE2 binding of each barcoded variant at each titration concentration on the basis of its distribution of counts across FACS bins calculated as a simple mean[60], as described previously[12]. We determined the apparent binding constant $K_{D,app}$ describing the affinity of each barcoded variant for each ACE2 along with free parameters $a$ (titration response range) and $b$ (titration curve baseline) with nonlinear least-squares regression using the standard non-cooperative Hill equation relating the mean bin response variable to the ACE2 labelling concentration:

$$\text{bin} = a \times [\text{ACE2}]/\left([\text{ACE2}] + K_{D,app}\right) + b$$

The measured mean bin value at a given ACE2 concentration was excluded from a variant's curve fit if fewer than 10 counts were observed across the four FACS bins at that concentration. Individual concentration points were also excluded from the curve fit if they demonstrated evidence of bimodality (>40% of counts of a barcode were found in each of two non-consecutive bins 1 + 3 or 2 + 4, or >20% of counts of a barcode were found in each of the boundary bins 1 + 4). To avoid errant fits, we constrained the fit baseline parameter $b$ to be between 1 and 1.5, the response parameter $a$ to be between 2 and 3, and the $K_{D,app}$ parameter to be between $1 \times 10^{-15}$ and $1 \times 10^{-5}$. The fit for a barcoded variant was discarded if the average count across all sample concentrations was below 10, or if >20% of sample concentrations were missing due to counts below 10. We also discarded curve fits in cases in which the normalized mean square residual (residuals normalized from 0 to 1 relative to the fit response parameter $a$) is >10× the median normalized mean square residual across all titrations with all ACE2s. $K_{D,app}$ binding constants were expressed as $-\log_{10}(K_{D,app})$, where higher values indicate higher-affinity binding. The full pipeline for computing per-barcode binding affinities is described at GitHub (https://github.com/jbloomlab/SARSr-CoV_homolog_survey/blob/master/results/summary/compute_binding_Kd.md).

To derive our final measurements we collapsed measurements across internally replicated barcodes representing each RBD genotype. For each RBD genotype, we discarded the top and bottom 5% (expression measurements) or 2.5% (titration affinities) of per-barcode measurements, and computed the mean value across the remaining barcodes within each library. The correlations in these barcode-averaged measurements between independently barcoded and assayed library replicates are shown in Extended Data Fig. 2g. Final measurements were determined as the mean of the barcode-collapsed mean measurements from each replicate. The total number of barcodes collapsed into these final measurements from both replicates is shown in the histograms in Extended Data Fig. 2f. Final measurements for an RBD genotype were discarded if the RBD genotype was not sampled with at least one non-filtered barcode in each replicate, or sampled with at least five non-filtered barcodes in a single replicate. The full pipeline for barcode collapsing is described at GitHub (https://github.com/jbloomlab/SARSr-CoV_homolog_survey/blob/master/results/summary/barcode_to_genotype_phenotypes.md). The final processed measurements of expression and ACE2 binding for parental and mutant RBDs can be found at GitHub (https://github.com/jbloomlab/SARSr-CoV_homolog_survey/blob/master/results/final_variant_scores/wt_variant_scores.csv and https://github.com/jbloomlab/SARSr-CoV_homolog_survey/blob/master/results/final_variant_scores/mut_variant_scores.csv).

### Isogenic ACE2-binding assays
For RBDs assayed subsequent to library experiments (Fig. 4 and Extended Data Figs. 4, 6f and 10e), RBDs were cloned as isogenic stocks into the 2649 plasmid, sequence verified and transformed individually into yeast using the LiAc/ssDNA transformation method[62]. Cultures were induced for RBD expression and labelled across ACE2 concentration series as described above in V-bottom 96-well plates with 0.067 $OD_{600}$ yeast per well. ACE2 labelling of RBD$^+$ cells was measured using the BD LSRFortessa X50 flow cytometer and data were processed using FlowJo (v.10). Binding curves of PE (ACE2) mean fluorescence intensity versus ACE2 labelling concentration were fit as above, with the inclusion of a Hill coefficient slope parameter $n$.

### Transient expression of *R. affinis* and *R. sinicus* ACE2–Fc
The *R. affinis* 787 (GenBank: QMQ39222.1), *R. affinis* 9479 (GenBank: QMQ39227.1), *R. sinicus* 1446 (GenBank: QMQ39213.1), *R. sinicus* WJ1 (GenBank: QMQ39206.1), *R. sinicus* GQ262791 (GenBank: ACT66275.1), *R. sinicus* 3364 (GenBank: QMQ39219.1), *R. sinicus* WJ4 (GenBank: QMQ39200.1), *R. sinicus* 1438 (GenBank: QMQ39203.1), *R. sinicus* 1434 (GenBank: QMQ39216.1) and *R. sinicus* 3358 (GenBank: QMQ39212.1) ACE2 ectodomains constructs were synthesized by GenScript and placed into a pCMV plasmid. The domain boundaries for the ectodomain are residues 19–615. The native signal tag was identified using SignalP-5.0 (residues 1–18) and replaced with an N-terminal mu-phosphatase signal peptide. These constructs were then fused to a sequence encoding a thrombin cleavage site and a human Fc fragment at the C-terminus. All ACE2–Fc constructs were produced in Expi293F cells (Thermo Fisher Scientific, A14527) in Gibco Expi293 Expression Medium at 37 °C in a humidified 8% $CO_2$ incubator rotating at 130 rpm. The cultures were transfected using PEI-25K (Polyscience) with cells grown to a density of 3 million cells per ml and cultivated for 4–5 days. Proteins were purified from clarified supernatants using a 1 ml HiTrap Protein A HP affinity column (Cytiva), concentrated and flash-frozen in 1× PBS, pH 7.4 (10 mM $Na_2HPO_4$, 1.8 mM $KH_2PO_4$, 2.7 mM KCl, 137 mM NaCl). Cell lines were not authenticated or tested for mycoplasma contamination.

### Transient expression of BtKY72 parental and mutant RBDs
BtKY72 RBD construct (BtKY72 S residues 318–520) was synthesized by GenScript into a CMVR plasmid with an N-terminal mu-phosphatase signal peptide and a C-terminal hexa-histidine tag (-HHHHHHHH) joined by a short linker (-GGSS) to an Avi tag (-GLNDIFEAQKIEWHE). BtKY72 mutant constructs T498W (BtKY72 S residue 487) and K493Y/T498W (BtKY72 S residue 482/487) were subcloned by GenScript from the BtKY72 RBD construct. BtKY72 and BtKY72 mutant RBD constructs were produced in Expi293F cells in Gibco Expi293 Expression Medium at 37 °C in a humidified 8% $CO_2$ incubator rotating at 130 rpm. The cultures were transfected using PEI-25K with cells grown to a density of 3 million cells per ml and cultivated for 3–5 days. Proteins were purified from clarified supernatants using a 1 ml HisTrap HP affinity column (Cytiva), concentrated and then biotinylated using a commercial BirA kit (Avidity). Proteins were then purified from the BirA enzyme by affinity purification using a 1 ml HisTrap HP affinity column (Cytiva), concentrated and flash-frozen in 1× PBS, pH 7.4. Cell lines were not authenticated or tested for mycoplasma contamination.

### BLI analysis
Assays were performed on an Octet Red (Forte Bio) instrument at 30 °C with shaking at 1,000 rpm. Streptavidin biosensors were hydrated in water for 10 min before incubation for 60 s in 10× kinetics buffer (undiluted). Biotinylated RBDs were loaded at 5–10 µg ml$^{-1}$ in 10× kinetics buffer for 100–600 s before baseline equilibration for 120 s in 10× kinetics

buffer. Association of ACE2–Fc (dimeric) was performed at 1 μM in 10× kinetics buffer. These data were baseline-subtracted. The experiments were performed with three separate purification batches of BtKY72 RBDs. All RBDs were immobilized to identical levels, that is, 1 nm shift. The data were plotted in GraphPad Prism and a representative plot is shown.

## Generation of VSV pseudovirus

The BtKY72 S construct was synthesized by GenScript and cloned into an HDM plasmid with a C-terminal 3× Flag tag. The BtKY72 mutant S constructs T498W (BtKY72 S residue 487) and K493Y/T498W (BtKY72 S residue 482/487) were subcloned by GenScript from the BtKY72 S construct. Pseudotyped VSV particles were prepared using HEK293T (ATCC CRL-11268) cells seeded into 10 cm dishes. HEK293T cells were transfected using Lipofectamine 2000 (Life Technologies) with a $S$-encoding plasmid in Opti-MEM transfection medium and incubated for 5 h at 37 °C with 8% $CO_2$ supplemented with DMEM containing 10% FBS. One day after transfection, cells were infected with VSV (G*ΔG-luciferase) and, after 2 h, infected cells were washed five times with DMEM before adding medium supplemented with anti-VSV G antibodies (I1-mouse hybridoma supernatant diluted 1:40, ATCC CRL-2700). Pseudotyped particles were collected 18–24 h after inoculation, clarified from cellular debris by centrifugation at 3,000$g$ for 10 min, concentrated 100× using a 100 MWCO membrane for 10 min at 3,000 rpm and frozen at −80 °C. Mock pseudotyped VSV pseudovirus was generated as above but in the absence of $S$. Cell lines were not authenticated or tested for mycoplasma contamination.

## VSV pseudovirus entry assays

HEK293T cells (ATCC CRL-11268) and HEK293T cells with stable transfection of human ACE2 (ref. [63]) were cultured in 10% FBS, 1% penicillin–streptomycin DMEM at 37 °C in a humidified 8% $CO_2$ incubator. Cells were plated into poly-lysine-coated 96-well plates. For *R. affinis* ACE2 entry, transient transfection of *R. affinis* ACE2 in HEK293T cells was performed 36–48 h before infection using Lipofectamine 2000 (Life Technologies) and an HDM plasmid containing full length *R. affinis* ACE2 (synthesized by GenScript) in Opti-MEM. After 5 h incubation at 37 °C in a humidified 8% $CO_2$ incubator, DMEM with 10% FBS was added and cells were incubated at 37 °C in a humidified 8% $CO_2$ incubator for 36–48 h. Cell lines were not authenticated or tested for mycoplasma contamination.

Immediately before infection, HEK293T cells with stable expression of human *ACE2*, transient expression of *R. affinis ACE2* or not transduced to express *ACE2* were washed once with DMEM, then plated with normalized pseudovirus in DMEM. Infection in DMEM was performed with cells between 60–80% confluence (human ACE2-293T) or between 80–90% confluence (*R. affinis* ACE2-293T) for 2.5 h before adding FBS and penicillin–streptomycin to final concentrations of 10% and 1%, respectively. After 24 h of infection, One-Glo-EX (Promega) was added to the cells and incubated in the dark for 5 min before reading on a Synergy H1 Hybrid Multi-Mode plate reader (Biotek). Normalized cell entry levels of pseudovirus generated on different days (biological replicates) were plotted in GraphPad Prism as individual points, and average cell entry across biological replicates was calculated as the geometric mean.

BtKY72 S parental and mutant pseudoviral particle inputs for the above cell entry assays were normalized to spike incorporation quantified using western blotting. Detection of S was performed using mouse monoclonal anti-Flag M2 antibodies (Sigma-Aldrich, F3165) and Alexa Fluor 680 AffiniPure Goat Anti-Mouse IgG, light chain specific (Jackson ImmunoResearch Labs, 115-625-174). Detection of the VSV backbone was performed using anti-VSV-M [23H12] antibodies (Kerafast, EB0011) and Alexa Fluor 680 AffiniPure Goat Anti-Mouse IgG, light chain specific (Jackson ImmunoResearch Labs 115-625-174). A representative blot is shown in Extended Data Fig. 3c. Expression of the *R. affinis ACE2* alleles was not quantified or normalized.

## Biosafety considerations

We characterized the human ACE2 binding of sarbecovirus RBDs and identified point mutants that increase the affinity of some RBDs. This work includes identifying sarbecovirus RBDs from outside southeast Asia that can naturally bind to human ACE2 (Khosta-2 RBD from Russia) or adapt to bind to human ACE2 with just a few mutations (BtKY72 RBD from Kenya). We verified this latter finding using non-replicative spike-pseudotyped VSV particles. None of our experiments pose a biosafety risk, as they involve only RBD protein (purified or expressed in yeast) or non-replicative pseudotyped VSV viral particles, and not live virus. However, it is possible that another researcher could perform experiments on actual sarbecoviruses with RBDs such as the ones we described, and such experiments could pose a risk. Against that possible information misuse, we weigh the following benefits of the information conveyed by our study: (1) as stated in the concluding paragraph of the Discussion, we used safe methods to highlight the need for care when sampling sarbecoviruses including those from outside southeast Asia; (2) we identified a broader swath of spike proteins that should be included in biochemical studies to engineer countermeasures (such as broad antibodies[64,65] or stabilized spike immunogens); (3) we characterized mutations that could enable safer mouse-adapted laboratory strains with reduced human ACE2 affinity (Extended Data Fig. 8c); (4) we provide data that can improve sequence-based phenotypic predictions. We emphasize that our research indicates that live-virus experiments with any new sarbecovirus should involve careful consideration of risks, as human ACE2 binding may be widespread. The actual ability of a sarbecovirus to infect humans will depend not only on its ACE2 affinity, but also other properties including proteolytic activation of the spike protein[66], innate immunity and other poorly understood factors.

## Reporting summary

Further information on research design is available in the Nature Research Reporting Summary linked to this paper.

## Data availability

PacBio CCSs are available from the NCBI SRA, BioSample SAMN18316101. Illumina sequences for barcode counting are available from the NCBI SRA, BioSample SAMN20174027. A table of measurements of ACE2 binding and expression for all parental RBDs is available at GitHub (https://github.com/jbloomlab/SARSr-CoV_homolog_survey/blob/master/results/final_variant_scores/wt_variant_scores.csv). A table of measurements of ACE2 binding and expression for all single mutant RBDs is available at GitHub (https://github.com/jbloomlab/SARSr-CoV_homolog_survey/blob/master/results/final_variant_scores/mut_variant_scores.csv). For bioinformatics analyses, tables of all virus names, species and location of sampling, and sequence accessions (NCBI GenBank or GISAID) or citations are provided at GitHub (https://github.com/jbloomlab/SARSr-CoV_homolog_survey/blob/master/RBD_ASR/RBD_accessions.csv).

## Code availability

All code for data analysis is available at GitHub (https://github.com/jbloomlab/SARSr-CoV_homolog_survey). A summary of the computational pipeline and links to individual notebooks detailing steps of analysis is available at GitHub (https://github.com/jbloomlab/SARSr-CoV_homolog_survey/blob/master/results/summary/summary.md).

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

**Acknowledgements** We thank the staff at the Fred Hutch Flow Cytometry and Genomics facilities and the Scientific Computing group supported by ORIP grant S10OD028685; Z. Shi for sharing *R. sinicus* and *R. affinis* ACE2 sequences from a preprint before formal publication; H. Tani for providing the reagents necessary for preparing VSV pseudotyped viruses; S. Goldstein and A. Chan for discussions; and all sequence contributors, including contributors to GISAID (https://github.com/jbloomlab/SARSr-CoV_homolog_survey/tree/master/RBD_ASR/gisaid). This study was supported by the National Institute of Allergy and Infectious Diseases (DP1AI158186 and HHSN272201700059C to D.V., and R01AI141707 to J.D.B.), the National Institute of General Medical Sciences (R01GM120553 to D.V., and 5T32GM008268-32 to S.K.Z.), a Pew Biomedical Scholars Award (to D.V.), Investigators in the Pathogenesis of Infectious Disease Awards from the Burroughs Wellcome Fund (to D.V. and J.D.B.), Fast Grants (to D.V.), the Bill & Melinda Gates Foundation (OPP1156262 to D.V., and INV-004949 to J.D.B.) and the Russian Foundation for Basic Research (20-04-60154 'An analysis of genetic diversity of zoonotic viruses in Russian populations of bats and rodents' to S.A.). T.N.S. is an HHMI Fellow of the Damon Runyon Cancer Research Foundation. D.V. and J.D.B. are investigators of the Howard Hughes Medical Institute.

**Author contributions** Conceptualization: T.N.S., S.K.Z., D.V. and J.D.B. Phylogenetics: T.N.S. Yeast-display methodology: T.N.S. and A.J.G. Yeast-display experiments: T.N.S. BLI measurements: S.K.Z. Pseudovirus entry assays: S.K.Z. and A.C.W. Data analysis: T.N.S. Khosta-1 and Khosta-2 sampling and sequencing: S.A. Supervision: D.V. and J.D.B. Writing—original draft: T.N.S., S.K.Z., D.V. and J.D.B. All of the authors read and edited the manuscript.

**Competing interests** J.D.B. consults for Moderna on viral evolution and epidemiology and Flagship Labs 77 on deep mutational scanning. J.D.B. may receive a share of IP revenue as an inventor on a Fred Hutchinson Cancer Research Center-optioned technology/patent (US Patent and Trademark Office application WO2020006494) related to deep mutational scanning of viral proteins. The Veesler laboratory (S.K.Z., A.C.W. and D.V.) has received an unrelated sponsored research agreement from Vir Biotechnology.

**Additional information**
**Correspondence and requests for materials** should be addressed to Tyler N. Starr, David Veesler or Jesse D. Bloom.

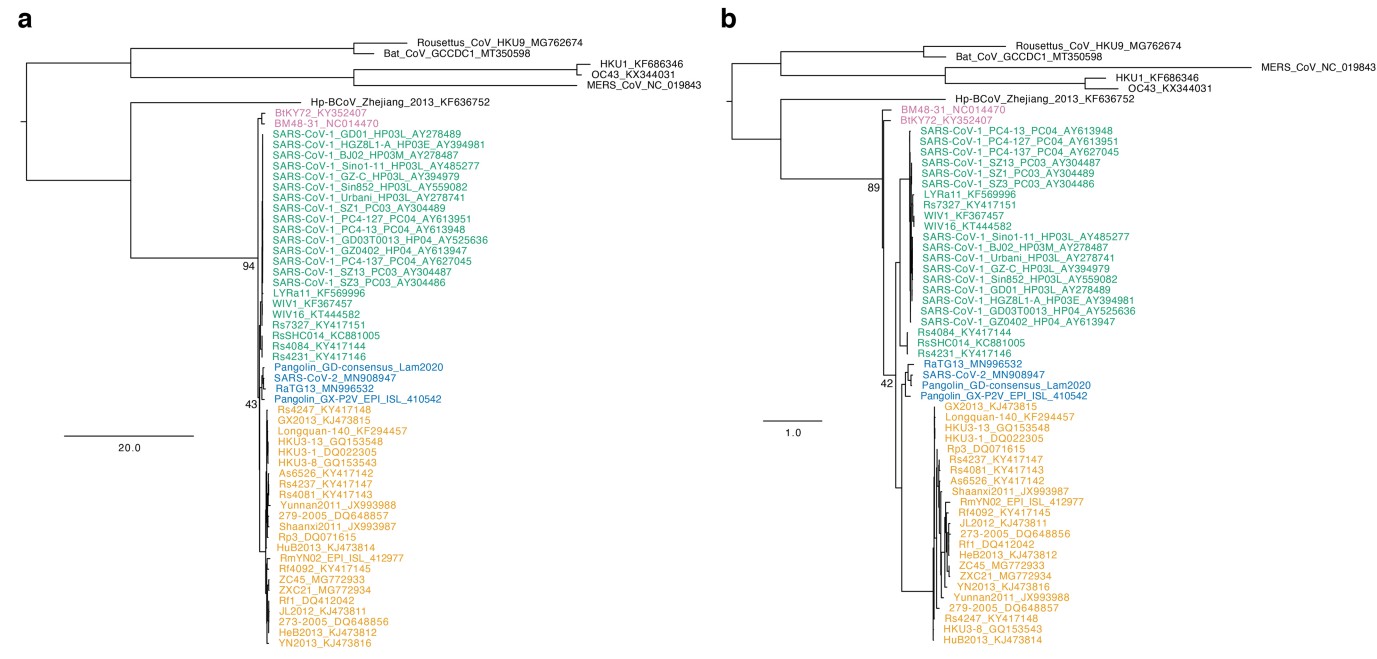

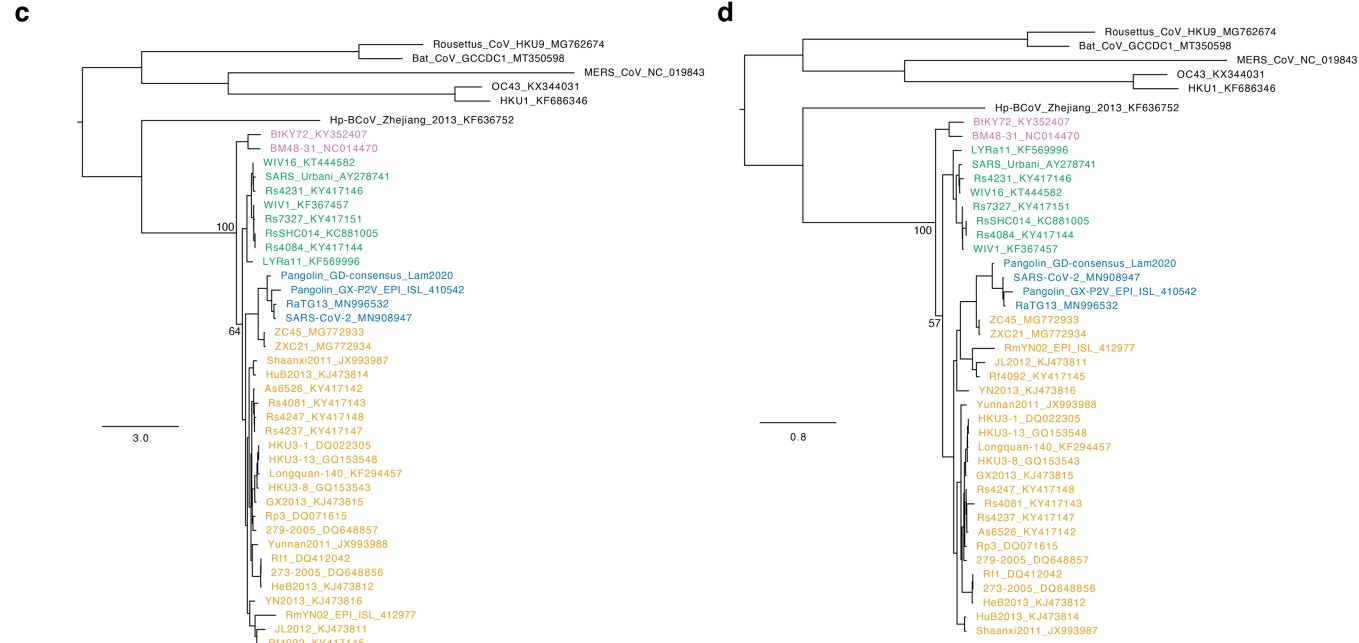

**Extended Data Fig. 1 | Robustness of the root of the sarbecovirus ingroup.** To establish robustness of our conclusion that the first sarbecovirus divergence is between sarbecoviruses from Africa and Europe and those from Asia, we inferred phylogenies based on alignments of RBD (SARS-CoV-2 spike residues N331-T531) (**a**,**b**) or the full spike gene (**c**,**d**) and nucleotide (**a**, **c**) or amino-acid (**b**, **d**) alignments and substitution models. In all four cases, the first sarbecovirus bipartition is placed between sarbecoviruses in Africa/Europe and those in Asia. The placement of the overall tree root is arbitrary with respect to the relationship among non-sarbecovirus outgroups, but this arbitrary placement does not impact the sarbecovirus ingroup rooting. The primary variations among trees includes a potential paraphyletic separation of BtKY72 and BM48–31 from Europe and Africa such that they do not form a monophyletic clade (**b**; also seen in Extended Data Fig. 10a-c), and variation in the relationships among the three Asia sarbecovirus clades (whose relationship is also inferred with a very low bootstrap support value in our primary phylogeny in Fig. 1a). Known recombination of RBDs with respect to other spike segments among viruses creates incongruencies between spike and RBD trees among Asian sarbecovirus lineages (e.g. ZC45 and ZXC21), though recombination has not been reported among the Africa and Europe spikes and those in Asia. Scale bar, expected number nucleotide or amino-acid substitutions per site. Node labels illustrate bootstrap support values for sarbecovirus and Asia sarbecovirus monophyly. Sequences colored by their RBD clade as in Fig. 1a.

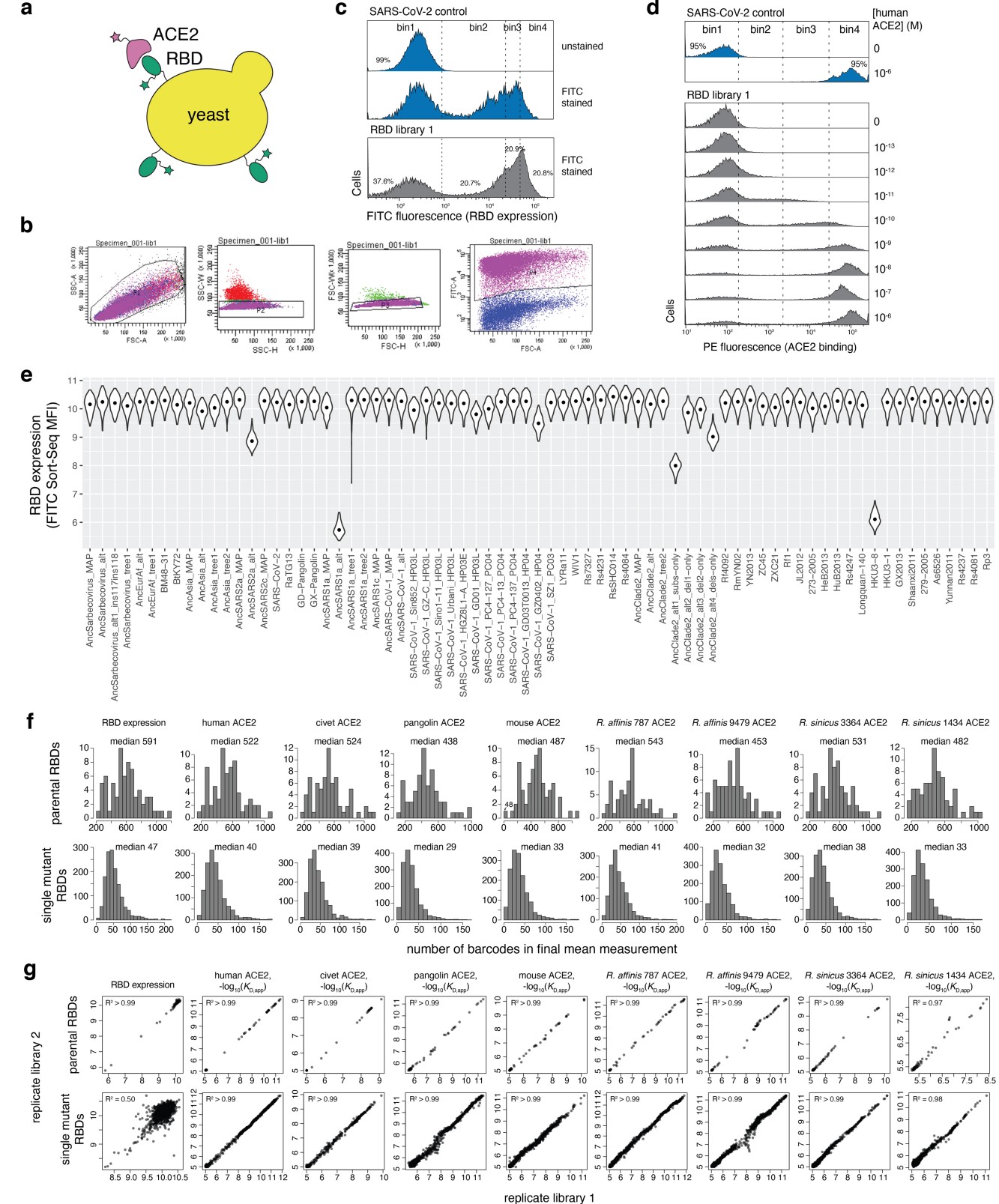

**Extended Data Fig. 2 | Experimental details of Sort-seq assays. a**, RBD yeast-surface display enables detection of folded RBD expression and ACE2 binding. **b**, Representative gating for single (SSC-A versus FSC-A, SSC-W versus SSC-H, and FSC-W versus FSC-W), RBD+ (FITC versus FSC-A) cells. **c**, Representative bins drawn on single cells for expression Sort-seq measurements. **d**, Representative bins drawn on single, RBD+ cells for ACE2 Tite-seq[12,67] measurements. **e**, Per-variant expression, shown as violin plots

across replicate barcodes representing each variant within the gene libraries. **f**, Number of distinct barcodes for each parental (top) or mutant (bottom) RBD genotype used in the determination of final pooled measurements across libraries. **g**, Correlation in measured phenotypes between independently assembled and barcoded gene library duplicates for parental (top) or mutant (bottom) RBD genotypes.

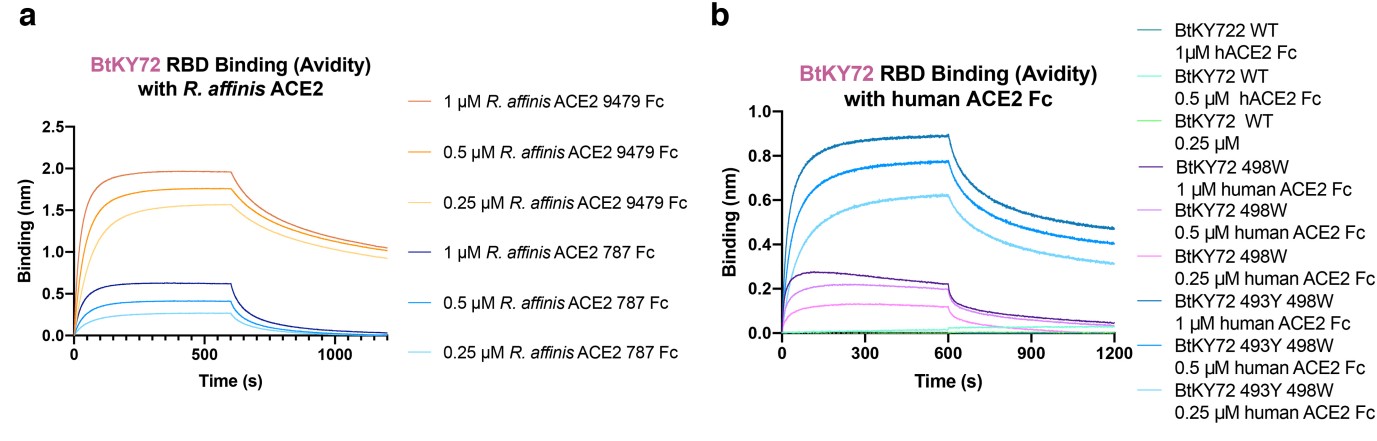

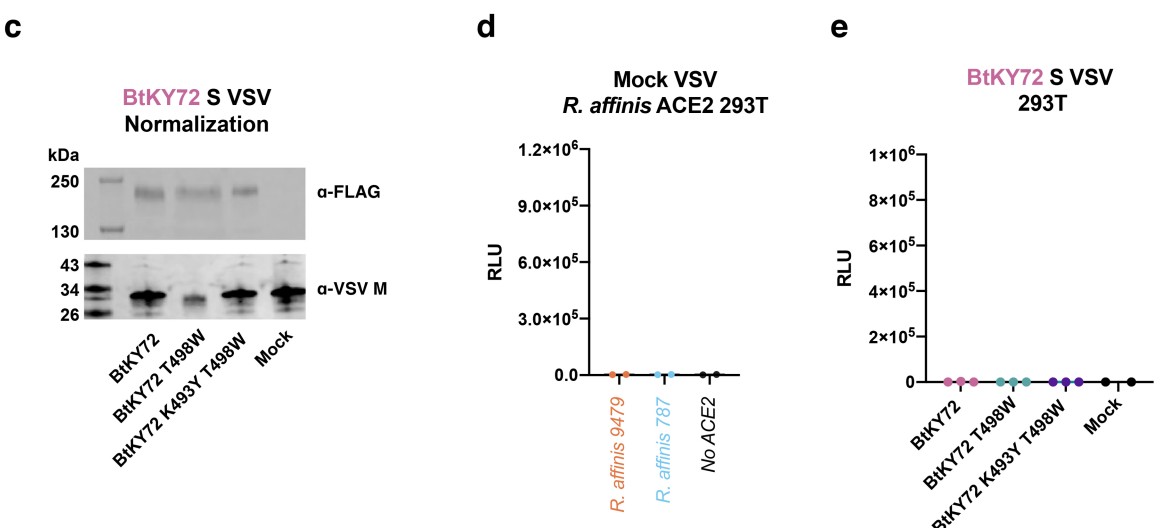

**Extended Data Fig. 3 | Normalization and controls for biolayer interferometry binding and pseudovirus entry assays. a,b** Biolayer interferometry binding analysis of a range of *R. affinis* ACE2-Fc (**a**) or human ACE2-Fc (**b**) concentrations to biotinylated BtKY72 RBD (parental or mutant) immobilized at the surface of streptavidin biosensors. **c,** Representative Western blots for quantification of spike incorporation into pseudoviral particles. Anti-FLAG (Sigma F3165) identifies incorporation of 3xFLAG-tagged spike, and anti-VSV-M (Kerafast EB0011) identifies level of VSV backbone. Viral inputs into cell entry assays were normalized across pseudoviral particles by S incorporation as determined in the anti-FLAG Western blot. Blot representative of biological duplicate generations of each pseudovirus. For gel source data, see Supplementary Fig. 1. **d,** Entry into *R. affinis* ACE2-expressing HEK293T cells by mock VSV particles produced in cells in which no spike gene was transfected. Each point represents the mean of technical triplicates for assays performed with independent preparation of pseudoviral particles (biological replicates). **e,** Entry of pseudoviral particles into HEK293T cells not transfected with any ACE2. Each point represents the mean of technical triplicates for assays performed with independent preparation of pseudoviral particles (biological replicates).

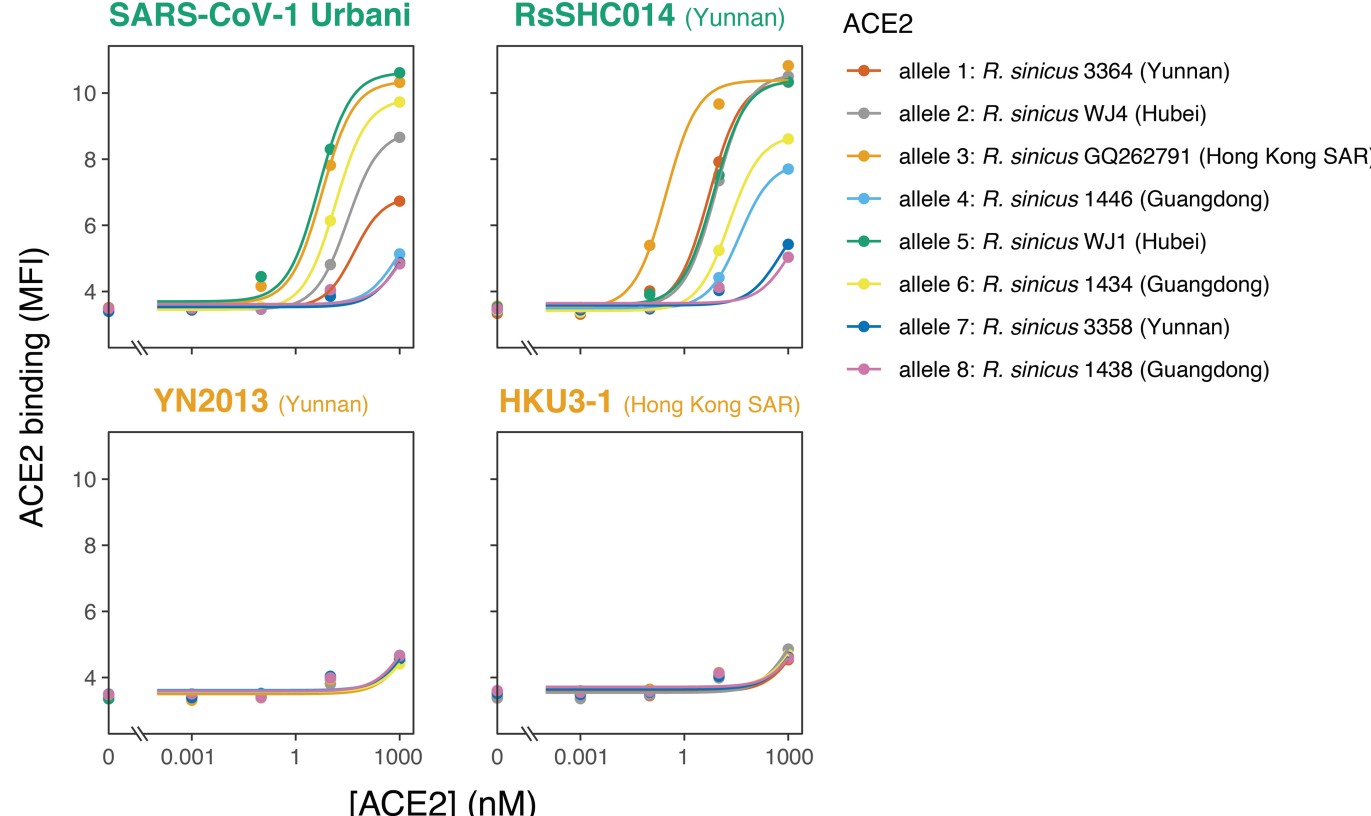

**Extended Data Fig. 4 | Clade 2 RBD binding to an expanded panel of *R. sinicus* ACE2 alleles.** Binding curves for Clade 1a (SARS-CoV-1 Urbani and RsSHC014) and Clade 2 (YN2013 and HKU3-1) sarbecovirus RBDs for 8 *R. sinicus* ACE2 alleles. Measurements performed with yeast-displayed RBDs and purified dimeric ACE2 proteins, measured by flow cytometry. Data from a single experimental replicate. Region of sampling for bat sarbecovirusess and *R. sinicus* ACE2 alleles are provided. RsSHC014, YN2013, and HKU3-1 were all sampled from *R. sinicus* bats.

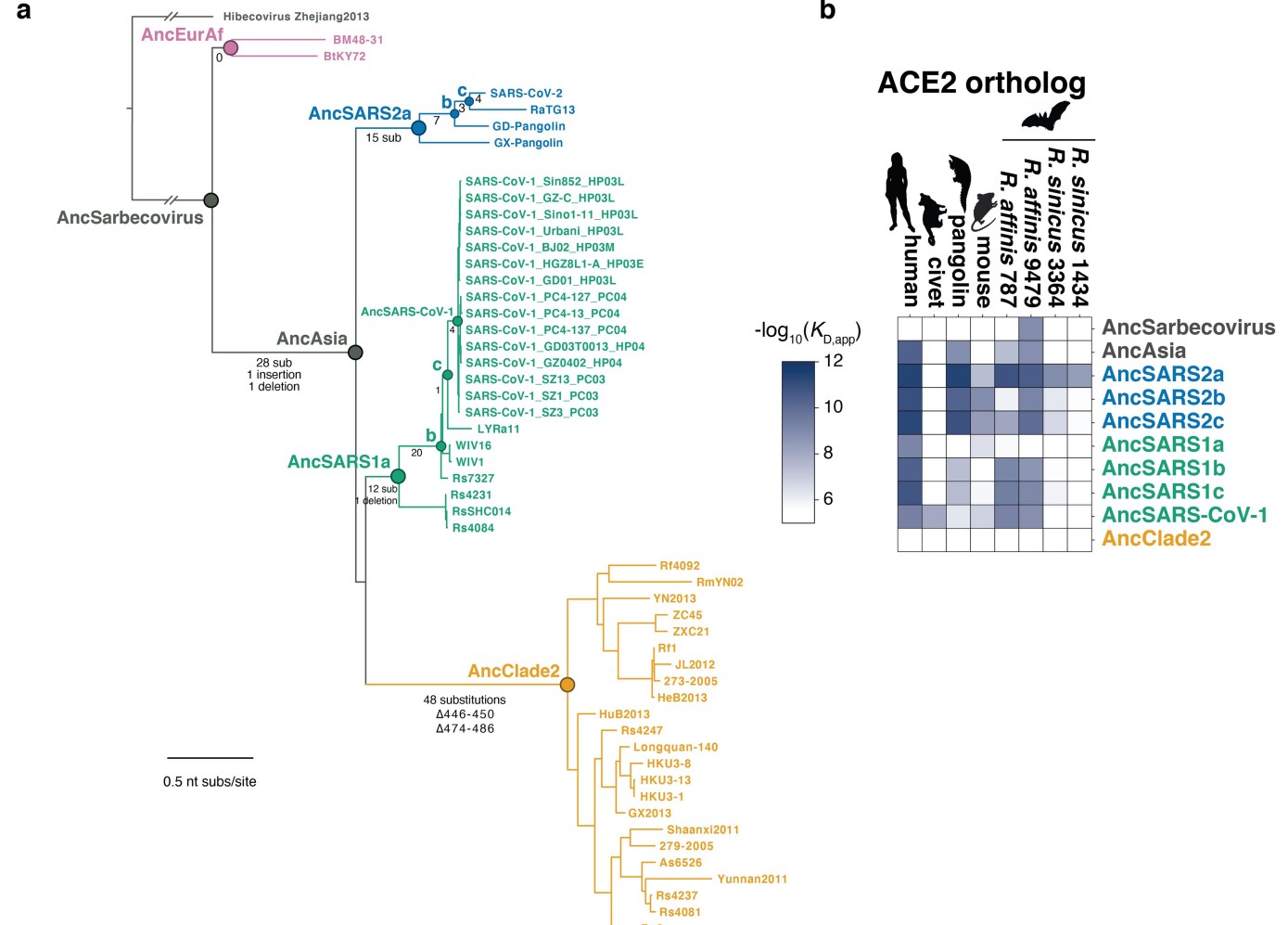

**Extended Data Fig. 5 | Full set of RBD ancestral sequence reconstructions. a**, Phylogeny with labelled nodes representing all ancestors tested, including nodes within the SARS-CoV-1 and SARS-CoV-2 clades leading to the human viruses. Branches are annotated with the number of amino-acid substitutions and indels that are inferred to have occurred along each branch. **b**, Phenotypes of all most plausible ancestral sequences (including repetition of the data represented in Fig. 2b).

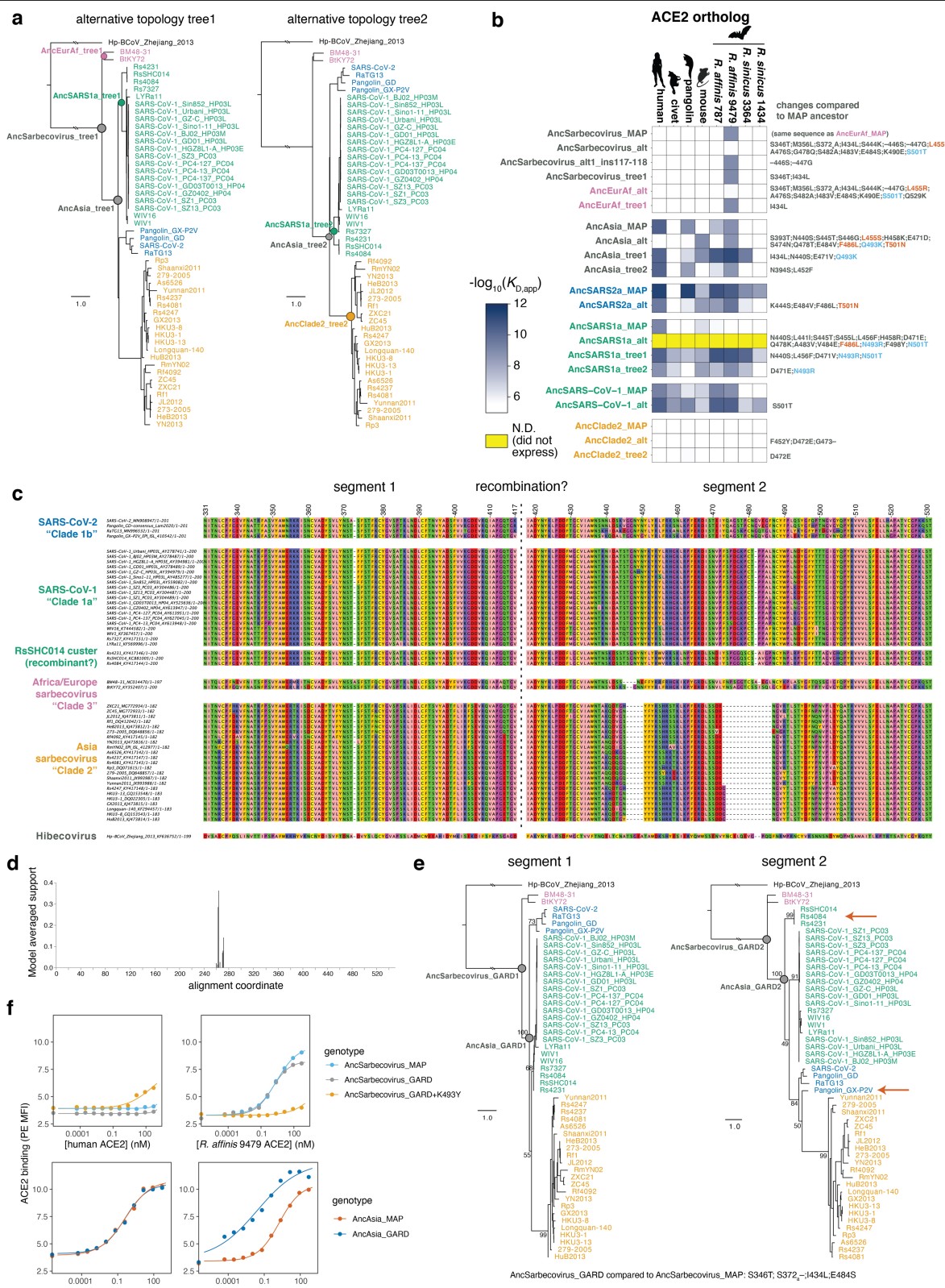

**Extended Data Fig. 6** | See next page for caption.

**Extended Data Fig. 6 | Robustness to uncertainties in ancestral reconstructions. a**, We performed ancestral sequence reconstructions on phylogenies constraining sister relationships between SARS-CoV-2 clade and clade 2 (tree1) or SARS-CoV-1 and SARS-CoV-2 clades (tree2) due to ambiguity in these relationships (Fig. 1a and Extended Data Fig. 1). **b**, ACE2 binding of alternative reconstructions. "Alt" ancestors incorporate all secondary reconstructed states with posterior probability > 0.2;[38] "tree1" and "tree2" ancestors are inferred on the constrained trees in (**a**); and "ins117-118" tests the ambiguity of an indel separate from the remaining substitutions in AncSarbecovirus_alt. Sequence differences are listed at right relative to the maximum a posteriori (MAP) ancestors from Fig. 2b and Extended Data Fig. 5b. Mutations are colored red if they were sufficient to abolish the ancestral phenotype and blue if they reinforced it (Extended Data Fig. 7). Dramatic changes to inferred ancestral phenotypes are mostly observed in the alt ancestors which are the most probabilistically distant, while the tree1 and tree2 alternatives generally recapitulate the MAP phenotypes. The exception is AncSARS1a, where the tree1 and tree2 alternatives better match what would be expected based on the descendent RBD phenotypes (Fig. 1b). **c**, RBD amino acid alignment, indicating a potential recombination breakpoint identified by GARD[56] (from underlying nucleotide sequence). **d**, Relative support values for possible recombination breakpoints. **e**, Phylogenies inferred for the putative non-recombinant RBD segments. Arrows point to key changes in the segment 2 sub-tree. Each change is supported by weak bootstrap support values, and this hypothesis introduces a non-parsimonious history with respect to an indel at position 482. We reconstructed AncSarbecovirus_GARD and AncAsia_GARD as concatenated segment 1 and 2 reconstructions. Mutations that distinguish the GARD and MAP ancestor are listed at bottom. **f**, Binding of GARD ancestors to human and *R. affinis* 9479 ACE2 was determined in isogenic yeast-display titrations.

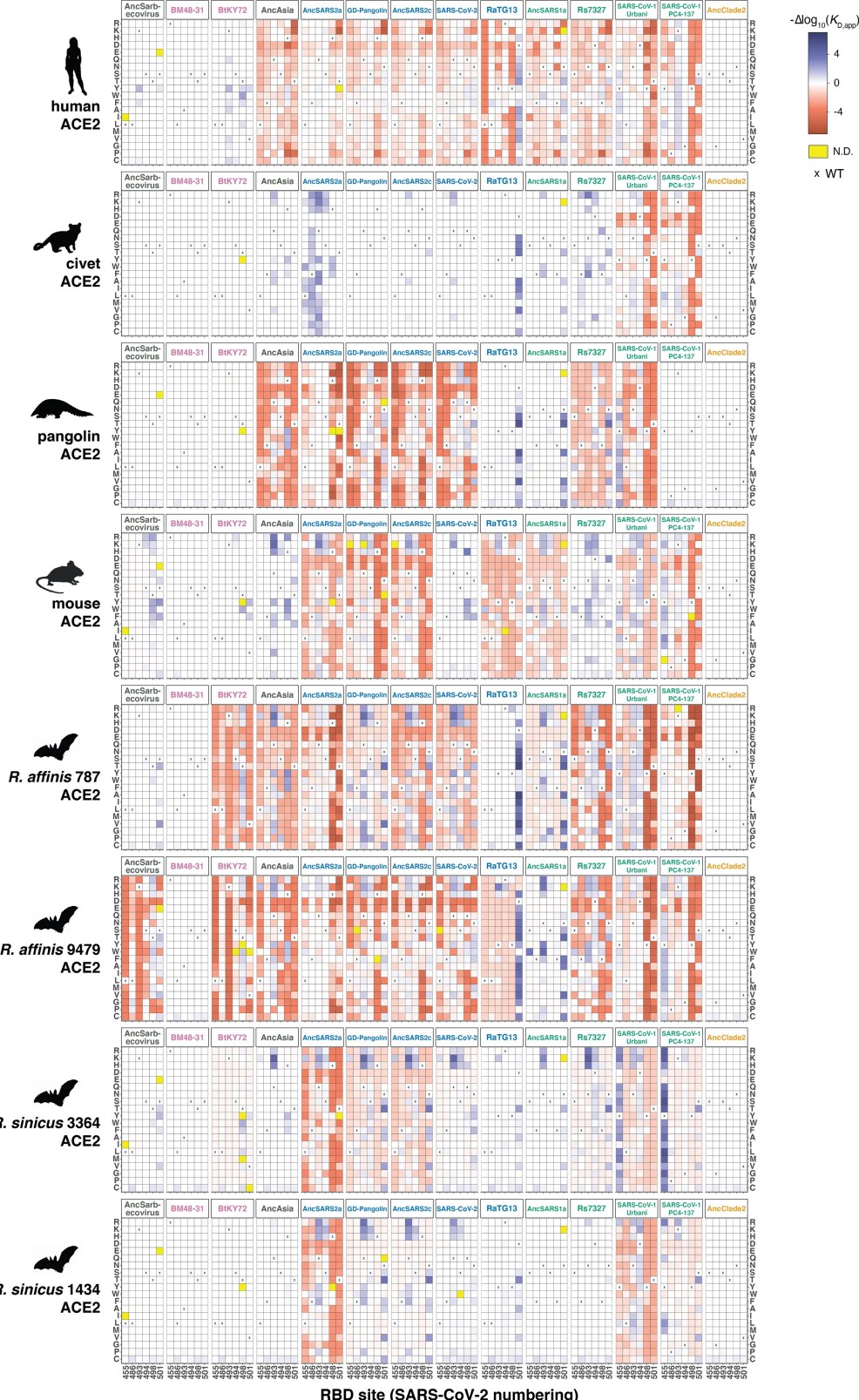

**Extended Data Fig. 7 | Binding of RBD single mutants to each ACE2.** Each heatmap square illustrates the change in binding caused by the indicated mutation at the indicated position (SARS-CoV-2 numbering), according to the color key shown on the upper-right. Yellow, mutations that were absent from the library or not sampled with sufficient depth in a particular experiment. x markers indicate the wildtype state at each position in each RBD background.

**a**

| virus | replication in lung of young BALB/c mouse | pathogenesis in young BALB/c mouse | notes | reference |
|---|---|---|---|---|
| SARS-CoV-1 Urbani infectious clone (i.c.) | efficient | none | N/A | Roberts et al. *PLOS Path* 2007 |
| SARS-CoV-1 Urbani MA15 | enhanced compared to SARS-CoV-1 i.c. | weight loss, lethality | six changes selected in serial passage; S Y436H key to pathogenesis | Roberts et al. *PLOS Path* 2007; Frieman et al. *JVI* 2012 |
| RsSHC014 i.c. | efficient, but lower than SARS-CoV-1 i.c. | none | N/A | Menachery et al. *Nat Med* 2015 |
| RsSHC014 S/ MA15 backbone | similar efficiency as SARS-CoV-1 MA15 | weight loss, no lethality | MA15 backbone minus S Y436H (RsSHC014 S) | Menachery et al. *Nat Med* 2015 |
| WIV1 i.c. | inefficient | none | N/A | Menachery et al. *PNAS* 2016 |
| WIV1 S/ MA15 backbone | inefficient | none | MA15 backbone minus S Y436H (WIV1 S) | Menachery et al. *PNAS* 2016 |
| SARS-CoV-2 WA1 i.c. | undetectable | none | N/A | Dinnon et al. *Nature* 2020 |
| SARS-CoV-2 WA1 MA | efficient | no weight loss or lethality; signs of impaired lung function | rationally designed, Q498Y+P499T | Dinnon et al. *Nature* 2020 |
| SARS-CoV-2 MASCp6 | efficient | no weight loss or lethality; signs of impaired lung function | five changes selected in serial passage; includes S N501Y | Gu et al. *Science* 2020 |
| SARS-CoV-2 WBP-1 | efficient | lethality | 1 deletion and 5 nonsyn mutations selected in serial passage; includes S Q493K and Q498H | Huang et al. *EBioMedicine* 2021 |
| SARS-CoV-2 B.1.1.7 | inefficient | none | alpha VOC; includes S N501Y | Montagutelli et al. *bioRxiv* 2021 |
| SARS-CoV-2 B.1.351 | efficient | none | beta VOC; includes S K417N, E484K, N501Y | Montagutelli et al. *bioRxiv* 2021 |
| SARS-CoV-2 P.1 | efficient | none | gamma VOC; includes S K417T, E484K, N501Y | Montagutelli et al. *bioRxiv* 2021 |

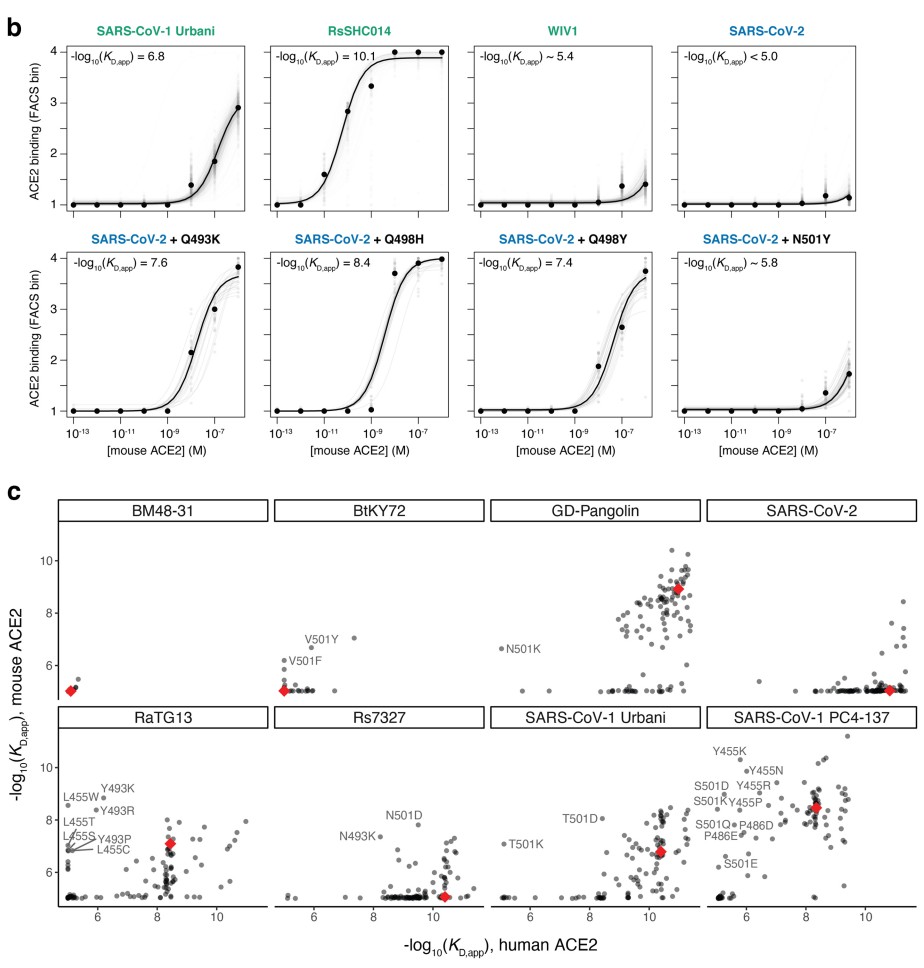

**Extended Data Fig. 8 |** See next page for caption.

**Extended Data Fig. 8 | Existing data on sarbecoviruses in mice, and affinities of RBDs and key mutants for mouse versus human ACE2.**
**a**, Summary of infectivity and pathogenesis of natural sarbecovirus and mouse-adapted strains from prior studies[31,42–44,68–71]. **b**, High-throughput titration curves for relevant genotypes from (**a**). Details as in Fig. 1d. Strength of binding to mouse ACE2 explains the infectivity and pathogenesis of SARS-CoV-1 Urbani and RsSHC014[31,43], relative to the weak or absent replication of WIV1[44] and SARS-CoV-2[42] in mice. Mutagenesis data explain the inefficient mouse infectivity of the SARS-CoV-2 B.1.1.7 isolate[71] which incorporates the N501Y RBD mutation, relative to the efficient replication of the mouse-adapted SARS-CoV-2 isolate containing Q498Y[42] or the pathogenic WBP-1 strain containing Q493K and Q498H[70]. **c**, An ideal mouse-adapted laboratory sarbecovirus strain would bind mouse ACE2 but not human ACE2 due to biosafety considerations. The large red points indicate the affinity of the parental RBD for human and mouse ACE2. The smaller black points indicate mutations, and key mutations that enhance binding to mouse versus human ACE2 are labelled (using SARS-CoV-2 numbering). Further mouse ACE2 specificity may be enabled via mutations at other positions not surveyed in our set of six positions.

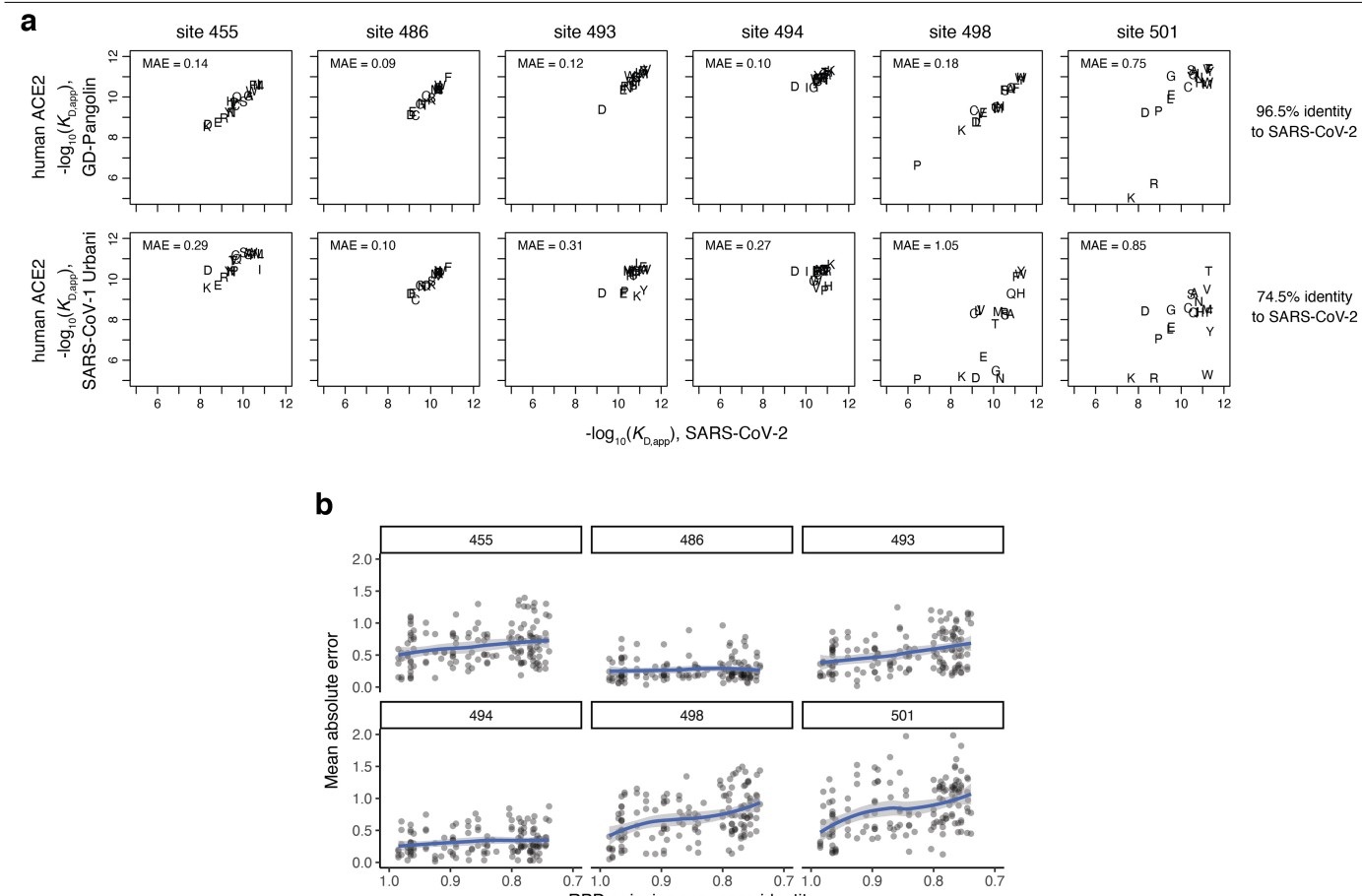

**Extended Data Fig. 9 | Epistasis and turnover in mutational effects.**
**a**, Example correlations in binding affinities for mutants in distinct RBD backgrounds at each site for human ACE2. Plots illustrate mutant avidities for human ACE2 and mean absolute error (residual) in the correlation for mutation measurements in GD-Pangolin (top) and SARS-CoV-1 Urbani (bottom) versus SARS-CoV-2. Plotting symbols indicate amino acid for each measurement.
**b**, Epistatic turnover in mutational effects across RBD backgrounds. Details as in Fig. 3g, but incorporating mutation effects among RBD pairs across all tested ACE2s. Blue line and shaded grey, LOESS mean and 95% CI trendline. See Extended Data Fig. 9b for analysis across all ACE2 orthologues.

## a
### RBD nucleotide

Hp-BCoV_Zhejiang_2013_KF636752
Khosta-2_MZ190138
RhGB01_MW719567
BM48-31_NC014470
BB9904_KR559017
PRD-0038_726045
PDF-2370_MT726044
BtKY72_KY352407
Khosta-1_MZ190137
RsYN04_EPI_ISL_1699444
SARS-CoV-2_MN908947
RshSTT182_EPI_ISL_852604
GD-Pangolin_Lam2020
RaTG13_MN996532
GX-Pangolin-P2V_EPI_ISL_410542
Rc-o319_LC556375
SARS-CoV-1_GZ-C_HP03L_AY394979
SARS-CoV-1_Sin852_HP03L_AY559082
SARS-CoV-1_Sino1-11_HP03L_AY485277
SARS-CoV-1_Urbani_HP03L_AY278741
SARS-CoV-1_BJ02_HP03M_AY278487
SARS-CoV-1_GD01_HP03L_AY278489
SARS-CoV-1_HGZ8L1-A_HP03E_AY394981
SARS-CoV-1_PC4-137_PC04_AY627045
SARS-CoV-1_GZ0402_HP04_AY613947
SARS-CoV-1_GD03T0013_HP04_AY525636
SARS-CoV-1_PC4-127_PC04_AY613951
SARS-CoV-1_PC4-13_PC04_AY613948
SARS-CoV-1_SZ13_PC03_AY304487
SARS-CoV-1_SZ1_PC03_AY304489
SARS-CoV-1_SZ3_PC03_AY304486
LYRa11_KF569996
Rs7327_KY417151
WIV16_KF367457
Rs4231_KY417146
RsSHC014_KC881005
Rs4084_KY417144
HuB2013_KJ473814
GX2013_KJ473815
Longquan-140_KF294457
HKU3-1_DQ022305
HKU3-13_GQ153548
HKU3-8_GQ153543
Rs4247_KY417148
Yunnan2011_JX993988
As6526_KY417142
Rs4237_KY417147
Rs4081_KY417143
Shaanxi2011_JX993987
279-2005_DQ648857
Rp3_DQ071615
YN2013_KJ473816
ZXC21_MG772934
ZC45_MG772933
HeB2013_KJ473812
JL2012_KJ473811
273-2005_DQ648856
Rf1_DQ412042
Rf4092_KY417145
RmYN02_EPI_ISL_412977
RacCS203_MW251308

1.0

## b
### RBD amino acid

Hp-BCoV_Zhejiang_2013_KF636752
RhGB01_MW719567
BB9904_KR559017
BM48-31_NC014470
BtKY72_KY352407
PDF-2370_MT726044
PRD-0038_726045
Khosta-2_MZ190138
Khosta-1_MZ190137
RsYN04_EPI_ISL_1699444
Shaanxi2011_JX993987
Yunnan2011_JX993988
RmYN02_EPI_ISL_412977
RacCS203_MW251308
Rf4092_KY417145
ZXC21_MG772934
ZC45_MG772933
Rf1_DQ412042
273-2005_DQ648856
HeB2013_KJ473812
Rp3_DQ071615
279-2005_DQ648857
HuB2013_KJ473814
HKU3-8_GQ153543
GX2013_KJ473815
Longquan-140_KF294457
HKU3-13_GQ153548
HKU3-1_DQ022305
Rs4237_KY417147
Rs4081_KY417143
As6526_KY417142
SARS-CoV-2_MN908947
GD-Pangolin_Lam2020
RshSTT182_EPI_ISL_852604
RaTG13_MN996532
GX-Pangolin-P2V_EPI_ISL_410542
Rc-o319_LC556375
SARS-CoV-1_PC4-127_PC04_AY613951
SARS-CoV-1_PC4-13_PC04_AY613948
SARS-CoV-1_PC4-137_PC04_AY627045
SARS-CoV-1_SZ1_PC03_AY304489
SARS-CoV-1_SZ3_PC03_AY304486
SARS-CoV-1_SZ13_PC03_AY304487
SARS-CoV-1_GZ-C_HP03L_AY394979
SARS-CoV-1_Sin852_HP03L_AY559082
SARS-CoV-1_BJ02_HP03M_AY278487
SARS-CoV-1_Urbani_HP03L_AY278741
SARS-CoV-1_Sino1-11_HP03L_AY485277
SARS-CoV-1_HGZ8L1-A_HP03E_AY394981
SARS-CoV-1_GD01_HP03L_AY278489
LYRa11_KF569996
WIV1_KF367457
Rs7327_KY417151
SARS-CoV-1_GD03T0013_HP04_AY525636
SARS-CoV-1_GZ0402_HP04_AY613947
Rs4084_KY417144
RsSHC014_KC881005
Rs4231_KY417146

1.0

## c
### spike nucleotide

Hp-BCoV_Zhejiang_2013_KF636752
Khosta-2_MZ190138
RhGB01_MW719567
BM48-31_NC014470
BB9904_KR559017
Khosta-1_MZ190137
BtKY72_KY352407
PDF-2370_MT726044
PRD-0038_726045
RsYN04_EPI_ISL_1699444
SARS-CoV-2_MN908947
RaTG13_MN996532
GX-Pangolin-P2V_EPI_ISL_410542
RshSTT182_EPI_ISL_852604
GD-Pangolin_Lam2020
ZXC21_MG772934
ZC45_MG772933
Rc-o319_LC556375
SARS_Urbani_AY278741
Rs4231_KY417146
WIV16_KT444582
WIV1_KF367457
Rs7327_KY417151
Rs4084_KY417144
RsSHC014_KC881005
LYRa11_KF569996
HuB2013_KJ473814
Shaanxi2011_JX993987
As6526_KY417142
Rs4237_KY417147
Rs4247_KY417148
279-2005_DQ648857
Rp3_DQ071615
GX2013_KJ473815
HKU3-8_GQ153543
Longquan-140_KF294457
HKU3-1_DQ022305
HKU3-13_GQ153548
Yunnan2011_JX993988
HeB2013_KJ473812
Rf1_DQ412042
273-2005_DQ648856
RacCS203_MW251308
RmYN02_EPI_ISL_412977
JL2012_KJ473811
Rf4092_KY417145
YN2013_KJ473816

1.0

## d
### spike amino acid

Hp-BCoV_Zhejiang_2013_KF636752
RhGB01_MW719567
Khosta-2_MZ190138
Khosta-1_MZ190137
BtKY72_KY352407
PRD-0038_726045
PDF-2370_MT726044
BB9904_KR559017
BM48-31_NC014470
RsYN04_EPI_ISL_1699444
SARS-CoV-2_MN908947
GX-Pangolin-P2V_EPI_ISL_410542
RaTG13_MN996532
GD-Pangolin_Lam2020
ZC45_MG772933
ZXC21_MG772934
RshSTT182_EPI_ISL_852604
Rc-o319_LC556375
SARS_Urbani_AY278741
WIV16_KT444582
Rs4231_KY417146
Rs7327_KY417151
Rs4084_KY417144
RsSHC014_KC881005
WIV1_KF367457
LYRa11_KF569996
Yunnan2011_JX993988
273-2005_DQ648857
Rf1_DQ412042
HeB2013_KJ473812
279-2005_DQ648857
GX2013_KJ473815
HKU3-8_GQ153543
Longquan-140_KF294457
HKU3-1_DQ022305
HKU3-13_GQ153548
Rs4247_KY417148
Rs4081_KY417143
As6526_KY417142
Rs4237_KY417147
Rp3_DQ071615
HuB2013_KJ473814
Shaanxi2011_JX993987
YN2013_KJ473816
JL2012_KJ473811
Rf4092_KY417145
RacCS203_MW251308
RmYN02_EPI_ISL_412977

0.5

## e

AncSarbecovirus_MAP compared to AncSarbecovirus_v2:
**Q340E;T346S;D360N;A372ₐS;G413A;L434I;R439N;I441L;K444S;Q445S;G446−;−448G;−449N;−450N;Y452F;A475P;−482s;P483I;S484E;E490K;V501S**

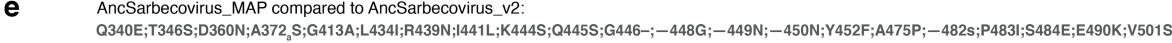

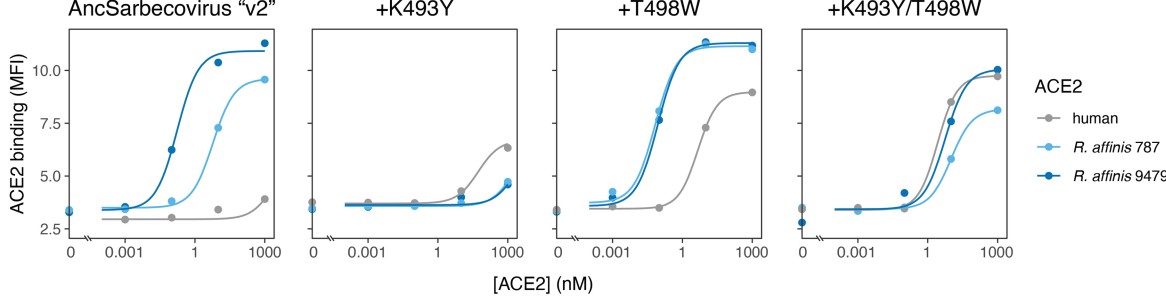

## Extended Data Fig. 10 | Robustness of rooting and AncSarbecovirus phenotype in a phylogeny incorporating newly reported sequences.

**a-d**, Phylogenetic inference with inclusion of newly reported sarbecovirus sequences (Fig. 4a). As in Extended Data Fig. 1, we infer phylogenies with RBD (**a**,**b**) and full spike alignments (**c**,**d**), both on nucleotide sequences (**a**,**c**) and translated amino acid (**b**,**d**) sequence alignments. The full set of outgroup betacoronavirus sequences shown in Extended Data Fig. 1 were also included in this tree inference but truncated from the display for visual clarity. The phylogeny in Fig. 4a is a constrained version of the RBD nucleotide tree from (**a**) where we constrained a monophyletic relationship among Africa/Europe sarbecoviruses due to uncertainty in the exact placement of the root within or relative ot the Africa/Europe sarbecovirus clade. **e**, ACE2 binding by parental RBD and candidate mutants in an updated AncSarbecovirus sequence ("v2") inferred from the phylogeny in Fig. 4a that incorporates many newly described sarbecovirus RBDs, including some in important new phylogenetic locations. The unconstrained tree in (**a**) leads to inference of an AncSarbecovirus sequence that is identical to Khosta-2 (which also binds ACE2). Sequence differences between the original MAP AncSarbecovirus and the "v2" reconstruction are listed at top. Measurements performed with yeast-displayed RBDs and purified dimeric ACE2 proteins, measured by flow cytometry. Data from a single experimental replicate.

# Reporting Summary

## Statistics

For all statistical analyses, confirm that the following items are present in the figure legend, table legend, main text, or Methods section.

| n/a | Confirmed | |
|---|---|---|
| ☐ | ☒ | The exact sample size (*n*) for each experimental group/condition, given as a discrete number and unit of measurement |
| ☐ | ☒ | A statement on whether measurements were taken from distinct samples or whether the same sample was measured repeatedly |
| ☒ | ☐ | The statistical test(s) used AND whether they are one- or two-sided<br>*Only common tests should be described solely by name; describe more complex techniques in the Methods section.* |
| ☒ | ☐ | A description of all covariates tested |
| ☒ | ☐ | A description of any assumptions or corrections, such as tests of normality and adjustment for multiple comparisons |
| ☐ | ☒ | A full description of the statistical parameters including central tendency (e.g. means) or other basic estimates (e.g. regression coefficient) AND variation (e.g. standard deviation) or associated estimates of uncertainty (e.g. confidence intervals) |
| ☒ | ☐ | For null hypothesis testing, the test statistic (e.g. *F*, *t*, *r*) with confidence intervals, effect sizes, degrees of freedom and *P* value noted<br>*Give P values as exact values whenever suitable.* |
| ☒ | ☐ | For Bayesian analysis, information on the choice of priors and Markov chain Monte Carlo settings |
| ☒ | ☐ | For hierarchical and complex designs, identification of the appropriate level for tests and full reporting of outcomes |
| ☒ | ☐ | Estimates of effect sizes (e.g. Cohen's *d*, Pearson's *r*), indicating how they were calculated |

*Our web collection on statistics for biologists contains articles on many of the points above.*

## Software and code

Policy information about availability of computer code

| Data collection | * Cell sorting experiments were operated using BD FACSDiva software (v. 8.0.2), and flow cytometry data processed in FlowJo (v. 10) |
|---|---|
| Data analysis | * phylogenetics and bioinformatics software includes mafft (v. 7.471), PAL2NAL (v. 14), RAxML (v. 8.2.12), FastML (v. 3.11) and GARD (v. 0.2)<br>* PacBio sequences were analyzed with ccs (v. 5.0.0) and alignparse (v. 0.1.6)<br>* Illumina sequences were processed with dms_variants (v. 0.8.5), and analyzed with fitdistrplus (v. 1.0.14)<br>* All custom code used for data analysis is available on GitHub: https://github.com/jbloomlab/SARSr-CoV_homolog_survey<br>* A summary of the computational pipeline and links to individual notebooks detailing steps of analysis is available on Github: https://github.com/jbloomlab/SARSr-CoV_homolog_survey/blob/master/results/summary/summary.md. Specific notebooks are listed below:<br>* All steps of bioinformatic analysis, including specific programmatic commands, alignments, raw data, and output files can be found on GitHub: https://github.com/jbloomlab/SARSr-CoV_homolog_survey/tree/master/RBD_ASR<br>* The PacBio CCS processing pipeline is available on GitHub: https://github.com/jbloomlab/SARSr-CoV_homolog_survey/blob/master/results/summary/process_ccs.md<br>* The full pipeline for computing per-barcode DMS expression values is described on GitHub: https://github.com/jbloomlab/SARSr-CoV_homolog_survey/blob/master/results/summary/compute_expression_meanF.md<br>* The full pipeline for computing per-barcode DMS binding affinities is described on GitHub: https://github.com/jbloomlab/SARSr-CoV_homolog_survey/blob/master/results/summary/compute_binding_Kd.md<br>* The full pipeline for barcode collapsing to final variant/mutant scores is described on GitHub: https://github.com/jbloomlab/SARSr-CoV_homolog_survey/blob/master/results/summary/barcode_to_genotype_phenotypes.md. |

For manuscripts utilizing custom algorithms or software that are central to the research but not yet described in published literature, software must be made available to editors and reviewers. We strongly encourage code deposition in a community repository (e.g. GitHub). See the Nature Portfolio guidelines for submitting code & software for further information.

## Data

Policy information about availability of data

All manuscripts must include a data availability statement. This statement should provide the following information, where applicable:

- Accession codes, unique identifiers, or web links for publicly available datasets
- A description of any restrictions on data availability
- For clinical datasets or third party data, please ensure that the statement adheres to our policy

* PacBio circular consensus sequences are available from the NCBI SRA, BioSample SAMN18316101
* Illumina sequences for barcode counting are available from the NCBI SRA, BioSample SAMN20174027
* Table of measurements of ACE2 binding and expression for all parental RBDs is available on GitHub: https://github.com/jbloomlab/SARSr-CoV_homolog_survey/blob/master/results/final_variant_scores/wt_variant_scores.csv
* Table of measurements of ACE2 binding and expression for all single mutant RBDs is available on GitHub: https://github.com/jbloomlab/SARSr-CoV_homolog_survey/blob/master/results/final_variant_scores/mut_variant_scores.csv
* All virus names, species and location of sampling, and sequence accessions (GenBank, GISAID) or citations are provided on GitHub: https://github.com/jbloomlab/SARSr-CoV_homolog_survey/blob/master/RBD_ASR/RBD_accessions.csv.

# Field-specific reporting

Please select the one below that is the best fit for your research. If you are not sure, read the appropriate sections before making your selection.

☒ Life sciences      ☐ Behavioural & social sciences      ☐ Ecological, evolutionary & environmental sciences

For a reference copy of the document with all sections, see nature.com/documents/nr-reporting-summary-flat.pdf

# Life sciences study design

All studies must disclose on these points even when the disclosure is negative.

| Sample size | No sample size determination was performed, as we were not performing statistical tests dependent on appropriate sample size determination |
|---|---|
| Data exclusions | No data were excluded from analyses |
| Replication | High-throughput titration measurements were replicated with two independently constructed gene libraries (Extended Data Fig. 2g). BLI binding assays were replicated in three batches of purified protein. Pseudovirus entry assays were replicated with two or three independent batches of pseudovirus generation. All experimental points are shown for DMS assays and pseudoviral entry assays, showing replication of results. Representative BLI traces are shown but were replicated, including when replicating under different sample concentrations (Extended Data Fig. 3a,b). |
| Randomization | Randomization was not performed. We conducted a standard survey of measurements across a panel of genotypes, which is not a study design that requires randomization |
| Blinding | Blinding was not performed in our study. High throughput titration experiments are conducted in massively parallel bulk experiments where there is no identifiability of individual variant genotypes, so blinding is not a relevant experimental attribute. |

# Reporting for specific materials, systems and methods

We require information from authors about some types of materials, experimental systems and methods used in many studies. Here, indicate whether each material, system or method listed is relevant to your study. If you are not sure if a list item applies to your research, read the appropriate section before selecting a response.

## Materials & experimental systems

| n/a | Involved in the study |
|---|---|
| ☐ | ☒ Antibodies |
| ☐ | ☒ Eukaryotic cell lines |
| ☒ | ☐ Palaeontology and archaeology |
| ☒ | ☐ Animals and other organisms |
| ☒ | ☐ Human research participants |
| ☒ | ☐ Clinical data |
| ☒ | ☐ Dual use research of concern |

## Methods

| n/a | Involved in the study |
|---|---|
| ☒ | ☐ ChIP-seq |
| ☐ | ☒ Flow cytometry |
| ☒ | ☐ MRI-based neuroimaging |

# Antibodies

| | |
|---|---|
| Antibodies used | FITC-conjutaged chicken anti-c-Myc (Immunology Consultants Lab, CYMC-45F); PE-conjugated streptavidin (ThermoFisher S866); iFluor-647-conjugated mouse anti-His (Genscript A01802); PE-conjugated goat anti-human IgG (Jackson ImmunoResearch Labs 109-115-098); mouse anti-VSV G (ATCC CRL-2700); Alexa Fluor 680-conjugated AffiniPure goat anti-mouse IgG (Jackson ImmunoResearch 115-625-174); mouse monoclonal anti-FLAG M2 antibody (Sigma F3165); Anti-VSV-M [23H12] antibody (Kerafast EB0011) |
| Validation | No validation was performed |

# Eukaryotic cell lines

Policy information about cell lines

| | |
|---|---|
| Cell line source(s) | * Expi293F: ThermoFisher A14527<br>* HEK293T: ATCC CRL-11268<br>* HEK293T-ACE2: Crawford, KHD et al. Protocol and reagents for pseudotyping lentiviral particles with SARS-CoV-2 spike protein for neutralization asasys. Viruses 12 (2020). |
| Authentication | Cell lines were not authenticated |
| Mycoplasma contamination | Cell lines were not tested for mycoplasma contamination |
| Commonly misidentified lines (See ICLAC register) | No commonly misidentified lines were used. |

# Flow Cytometry

## Plots

Confirm that:

☒ The axis labels state the marker and fluorochrome used (e.g. CD4-FITC).

☒ The axis scales are clearly visible. Include numbers along axes only for bottom left plot of group (a 'group' is an analysis of identical markers).

☒ All plots are contour plots with outliers or pseudocolor plots.

☒ A numerical value for number of cells or percentage (with statistics) is provided.

## Methodology

| | |
|---|---|
| Sample preparation | Yeast libraries expressing a library of sarbecovirus RBD variants on the cell surface were induced using standard culture techniques, as described in the Methods |
| Instrument | Sorting was conducted on a BD FACSAria II cell sorter. Flow cytometry analysis was conducted on a BD LSRFortessa X50 flow cytometer. |
| Software | Cell sorting experiments were operated using BD FACSDiva software (v. 8.0.2), and flow cytometry data processed in FlowJo (v. 10) |
| Cell population abundance | We were not sorting a specific target population, but rather partitioning all cells into encompassing bins on the basis of expression or ACE2 labeling, for downstream sequencing and reconstruction of per-variant labeling. |
| Gating strategy | Single cells were selected via FSC/SSC, FSC-W/FSC-A, and SSC-W/SSC-A gating. RBD-expressing cells were gated using a FITC/FSC gate. Single, RBD+ cells were sorted into bins of fluorescence on the basis of unlabeled or labeled control cells expressing the unmutated SARS-CoV-2 RBD. Representative gating schemes are illustrated in Extended Data Fig. 2b-d. |

☒ Tick this box to confirm that a figure exemplifying the gating strategy is provided in the Supplementary Information.

