## [Peer Review File · Nature]

Manuscript Title: ACE2 binding is an ancestral and evolvable trait of sarbecoviruses

Reviewer Comments & Author Rebuttals

Reviewer Reports on the Initial Version:

Referee #1

In this study, Starr et al. used yeast display with NGS to trace ACE2 binding across a diverse range of sarbecovirus RBDs and ACE2 orthologs. They also included the inferred ancestral sequences on the sarbecovirus RBD phylogeny to test ACE2 orthologs binding. Through this method, they proposed that ACE2 binding is an ancestral trait of sarbecovirus RBDs which was subsequently lost in some clades. By validation of BtKY72 binding to two *R. affinis* ACE2 alleles, they demonstrated for the first time that bat sarbecoviruses from outside Asia can bind ACE2. Site saturation mutagenesis libraries of six ACE2-RBD interaction residues revealed that single amino-acid mutations on many RBDs could enhance binding to new ACE2 orthologs, but this effect varies between viruses.

Major issues

1) Current *R. affinis* ACE2 sequences are all from China, but the *R. affinis* bats are also distributed in other countries, especially in South-East Asia. In this way, for Asia "clade 2" sarbecoviruses, the possibility of ACE2 usage is still not completely ruled out as not all *R. affinis* ACE2 are examined. Furthermore, the most geometrically distributed *Rhinolophus* bat is *R. ferrumequinum* which also hosts sarbecoviruses. It is not clear why other *Rhinolophus* bat ACE2 were not evaluated in this study? To be complete, it will be useful to also employ a mutagenesis library of ACE2 to provide the "other side" of the story.

2) The chosen 45 RBDs may not fully represent all ACE2-using and non-ACE2-using sarbecoviruses available. For example, Rc-o319 from Japan, BB9904 from Bulgaria, RshSTT182 from Cambodia, RacCS203 from Thailand were not examined for ACE2 orthologs binding. For Rc-o319 it also showed bat ACE2 binding, but not *R. affinis*. In this case, the completeness of the investigation is impaired and when it comes to the ancestral sequence reconstruction, these sequences should be included and a comparison should be made to see if any conclusions would change after such sequences are added.

3) For Figure 3, Extended Figure 6 and related results for ACE2 binding evolution, it is better to include the RsSHC014 clade. This clade is attract great interest in vaccine development as SARS-CoV-2 antibodies generated from the current vaccines were much less effective to neutralize this clade than the other sarbecoviruses. Furthermore, this RsSHC014 clade alone can bind both *R. affinis* and *R. sinicus* from modestly to strongly while other human and civet SARS-CoV-1, "Clade 1a" and SARS-CoV-2 "Clade 1b" did not show such a pattern. Would these suggest that the RsSHC014 clade was evolutionary different from SARS-CoV-1 ad SARS-CoV-2?

Minor points:

For ACE2 orthologs affinity, please clarify "no", "weak", "modest", and "strong" as the criteria is vague between different statements.

Line 53-55: How were the two distinct *R. affinis* and *R. sinicus* ACE2 alleles chosen? For each species, were the chosen two have the largest distance in the alignment of RBD-interaction residues? Please clarigy.

Line 67-68: "Binding to civet ACE2 was only detected within the SARS-CoV-1 clade whereas pangolin ACE2 binding is strongest within the SARS-CoV-2 clade". For pangolin ACE2, Sino1-11 in the SARS-CoV-1 clade also have very strong binding, if not the strongest?

Line 214 and Figure 4: Does the inclusion of new sarbecoviruses such as PDF-2370 and RsYN04 affect the ancestral sequence showed and discussed on Figure 3 and Extended Figure 5

Referee #2:

Tyler et al performed a comprehensive analysis of the ACE2 binding of the known sarbecovirus RBDs and their mutations by using the yeast-surface display platform, deep mutational scanning and high-throughput measurement of ACE2. The authors also reconstructed the ACE2 binding ancestor of all sarbecovirus RBDs based on one of two tested *R. affinis* ACE2 allele. The major conclusions include: 1) the human ACE2 binding is restricted to RBDs with SARS-CoV-2 and SARS-CoV-1 clade 1 and show different affinity among the tested RBDs. 2) Binding to civet ACE2 was only detected within the SARS-CoV-1 clade 1 whereas pangolin ACE2 binding is strongest within the SARS-CoV-2 clade. 3) Mouse ACE2 has a low or no binding affinity to the tested RBDs. 4) The two *R. sinicus* bat ACE2 alleles tested only interact with SARS-CoV-1 clade 1 isolates. No binding to these *R. sinicus* ACE2s was detected in the SARS-CoV-2 clade. Many SARS-CoV-2 and SARS-CoV-1 clade 1 RBDs bind to *rhinolophus affinis* ACE2. 5) For the first time authors detected the binding to a *rhinolophus affinis* ACE2 allele (discovered in China) by the RBD of a Kenya sarbecovirus BtKY72. 6) No ACE2 binding was detected for the RBDs of SARS-CoV-1 clade 2. 7) The ACE2 binding is an ancestral trait of the sarbecovirus and some of them (SARS-CoV-1 clade 2) lost the ACE2 binding during the evolution. 7) Most of single amino-acid mutations at the six key ACE2-contact positions improved sarbecovirus RBD binding to ACE2 orthologs except the European sarbecovirus and SARS-CoV-1 clade 2. Several single aa mutations enable the BtKY72 RBD to bind to human ACE2. These results suggest that the ACE2 binding of sarbecovirus RBDs is evaluable and show high plasticity.

Overall, this manuscript presents an up-to-date overview of the binding affinity between sarbecovirus RBD and ACE2 orthologs. It's important for the spillover risk assessment of the novel sarbecoviruses in terms of ACE2 receptor utilization. Although most of the conclusions are not novel to the specific research field, it's highly interesting to much broader readership considering that there are two sarbecoviruses which have caused emerging infectious diseases in the last 20 years.

Comments:

1. As the ACE2 utilization is an ancestral trait of sarbecovirus, it will be interesting to know if this feature is correlated with their evolutionary history. This reviewer suggests authors perform a molecule clock analysis using the RNA-dependent RNA polymerase of all known sarbecovirus sequences and compare the ancestor reconstruction tree by the ACE2 binding.
2. During the preparation of this manuscript, several publications have reported the ACE2 binding by the novel sarbecovirus RBDs. These novel strains have similar binding features as the Kenyan strain and bind to bat ACE2 not human ACE2. Authors please add these sequences in the alignment or phylogenetic trees if their binding analyses of the wildtype and mutations give too much work.
3. For the BLI assay in this manuscript, this reviewer would suggest use a serially concentration of ACE2 protein instead of one concentration three times, it will be more accurate to know if the binding affinity is stable under different concentrations.
4. Phylogenetic analysis using the RdRp sequence classifies the SARS-CoV-1 lineage into two clades, clade 1 utilize ACE2, clade 2 not. Authors please unify these terms throughout the text.

Referee #3

I am pleased to review this interesting and potentially important paper about using a high throughput technique involving expression of the viral spike proteins in yeast and binding with ACE2s from different species (human, some bat species and other mammals). Your experiments to measure the binding of spike proteins from a set of different sarbecoviruses to the ACE2 receptor have resulted in some highly useful data, and as you have used this and similar techniques before it is well described in the methods (but please see the likely comments from other reviewers on the detailed experimental methods, I am not specialist in that area).

What sets this work apart from other work on receptor binding in sarbecoviruses is the generation and measurement of spike sequences which are only inferred to exist (rather than examples of what has actually been isolated in nature). The methods of the phylogenetic reconstruction are as

I would expect (using more or less the software and models that I would expect) - firstly the tree is built, then a good evolutionary model is used for the ancestral state reconstruction; and alternative amino acids were also evaluated where there was uncertainty in the reconstructed ancestral sequences which is good.

Generally the paper is well written, and there is a lot of complex data to present. So it is much appreciated that you have made your code and results available on GitHub. This is a substantial body of code and results, and shows your commitment to open science. I notice that you stop short of trying to establish a binding prediction model for this paper, but you are already presenting a lot of work. I hope that the results (e.g. wt_variant_scores.csv and mut_variant_scores.csv etc) will be used by you and others in further analysis, e.g. for future viral fitness prediction models.

Minor comments

Phylogenetic methods lines 281-295: on reading this it seems that you have made the trees from only the nucleotides in the RBD region (about 600 nucleotides) ? Whilst this is enough to distinguish the main clades, it is probably contributing to the low bootstrap support on certain parts of the tree. Are the other regions of spike too divergent (or too many indels) to align well / or you are worried about recombination ? I see that you've tried it a few different ways, and also tried to assess recombination with GARD. Perhaps you can add little more clarification in this section ? (For SARS-CoV-2 and your earlier work the RBD region is approximately spike amino acids 330-350 ?, maybe just include some position numbers to give the general idea ?) But noted that you also say about the reconstructions, alternative reconstructions and robustness of phenotype in the main text.

Referee #4

This submission expands on the authors' previous Cell paper of 2020, "Deep mutational scanning of SARS-CoV-2....". In this submission, the focus is again on sarbecovirus receptor binding domains (RBDs) and their interaction with human and orthologous ACE2 receptors. Using high throughput RBD-ACE2 binding assays, the authors made several findings that are consistent with previous studies, and they also made some important original discoveries. A valuable new finding is that geographically isolated sarbecoviruses from Africa will use bat ACE2 for entry. Related to this finding are data demonstrating that sarbecovirus RBDs are "evolutionarily plastic", in that single aa substitutions markedly change orthologous ACE2 binding affinities. Together these findings point out that zoonotic sarbecovirus spillovers into humans might take place on continents beyond Asia.

The submission also highlights the utility of the high throughput RBD-ACE2 binding assays. The authors point out that complex RBD context makes sequence-based predictions of hACE2 binding nearly impossible - hence there are needs for the approaches illustrated in the submission. These approaches were shown once again to yield important information - a good example is with the analyses of RBD bindings to human and mouse ACE2, and preliminary identification of specifically mouse ACE2-binding RBDs for eventual engineering into "safe" mouse adapted viruses.

The study is very thorough and treats uncertainties carefully. There is support for a central argument that sarbecovirus utilization of ACE2 is an ancestral trait, from which ACE2 usage can be lost through deletion of RBD residues. Reconstruction of putative ancestral RBDs was considered with attention to inherent ambiguities and the results do support divergent, not convergent evolution of RBDs in Asia and Africa.

There are only minor suggestions for improvements: (1) there could be some mention of RBD evolution in the embeco and merbeco groups, so that readers appreciate that RBD plasticity extends beyond sarbecos and ACE2 receptors, (2) there could be some consideration of whether the clade 2 RBD deletions eliminate this domain's cell binding activity, with the NTDs then serving as principal RBDs (some literature may support this suggestion), (3) there could be some mention of possibly frequent sarbecovirus recombinations, which could generate ACE2 "gain" in clade 2

viruses (loss and gain of “ancestral” ACE2 binding might be ongoing and frequent through the process of discontinuous viral RNA replication).

Author Rebuttals to Initial Comments:

Referees' comments:

Referee #1:

In this study, Starr et al. used yeast display with NGS to trace ACE2 binding across a diverse range of sarbecovirus RBDs and ACE2 orthologs. They also included the inferred ancestral sequences on the sarbecovirus RBD phylogeny to test ACE2 orthologs binding. Through this method, they proposed that ACE2 binding is an ancestral trait of sarbecovirus RBDs which was subsequently lost in some clades. By validation of BtKY72 binding to two *R. affinis* ACE2 alleles, they demonstrated for the first time that bat sarbecoviruses from outside Asia can bind ACE2. Site saturation mutagenesis libraries of six ACE2-RBD interaction residues revealed that single amino-acid mutations on many RBDs could enhance binding to new ACE2 orthologs, but this effect varies between viruses.

Major issues

1) Current *R. affinis* ACE2 sequences are all from China, but the *R. affinis* bats are also distributed in other countries, especially in South-East Asia. In this way, for Asia “clade 2” sarbecoviruses, the possibility of ACE2 usage is still not completely ruled out as not all *R. affinis* ACE2 are examined. Furthermore, the most geometrically distributed *Rhinolophus* bat is *R. ferrumequinum* which also hosts sarbecoviruses. It is not clear why other *Rhinolophus* bat ACE2 were not evaluated in this study? To be complete, it will be useful to also employ a mutagenesis library of ACE2 to provide the “other side” of the story.

For the current manuscript, we did test all ACE2 RBD-interface configurations that have been described for *R. affinis* (there are only two), which were sampled from *R. affinis* bats in Yunnan and Hubei, China, where most of the RBD Clade 2 sarbecoviruses in our panel were sampled (8/23 from Yunnan, 4/23 from Hubei). However, the reviewer’s point stands that we only tested two alleles of *R. sinicus* ACE2, while 8 RBD-interface configurations have been described from *R. sinicus* bats sampled from Yunnan, Hubei, Guanxi, and Guangdong provinces and Hong Kong Special Administrative Region [HKSAR] (17/23 Clade 2 RBDs in our panel are from these regions).

To lend further support to the hypothesis that Clade 2 RBDs have lost ACE2 utilization, we tested binding of YN2013 and HKU3-1 RBDs, Clade 2 RBDs isolated from *R. sinicus* bats in Yunnan and HKSAR, respectively, to each of the 8 *R. sinicus* RBD-interface alleles. In contrast to SARS-CoV-1 Urbani and RsSHC014, a Clade 1a RBD also isolated from *R. sinicus* bats in Yunnan, YN2013 and

HKU3-1 RBD did not bind to any of the 8 *R. sinicus* ACE2 alleles. These results are now illustrated in Extended Data Fig. 4, and described in the relevant section of the main text. We also added more description of the provinces of sampling of Clade 2 RBDs and these *R. sinicus* and *R. affinis* ACE2 alleles in the main text, Extended Data Fig. 4, and in a table on GitHub linked in the Methods, to highlight their geographic concordance. We maintain conservative language in the manuscript as we can never rule out that we are missing other ACE2 orthologs that can be bound by Clade 2 RBDs unless all possible ACE2 sequences are known and tested, but we feel this experiment adds additional important weight to the idea that Clade 2 RBDs have lost ACE2 binding. We note that our other conclusions (that ACE2 binding is present and evolvable in the ancestral sarbecovirus and in viruses from Africa and Europe) are “positive” results that cannot be undone by further expansion of the ACE2 panel, and it is only this single “negative” result that could potentially be overturned with further ACE2 sampling.

It is true that we use a limited repertoire of *Rhinolophus* bat ACE2s—although our assay is high-throughput with respect to screening RBDs and RBD mutants, it does not scale efficiently to large numbers of ACE2s. Therefore, we selected the bat species (*R. affinis* and *R. sinicus*) with published ACE2 sequences that are known to host both ACE2-utilizing and non-utilizing sarbecoviruses, and other species related to cross-species transmission (human, civet, pangolin) or lab research (mouse). We tried to purify *R. ferrumequinum* ACE2 given the geographic breadth of this host species range, as the reviewer points out, but this protein failed to express using two different constructs that we initially designed. However, we fully agree on the importance of expanding our understanding of how corresponding diversification of bat ACE2s also contributes to RBD evolution, and this is a key focus of future work including development of new experimental tools to increase throughput of surveying ACE2 sequences.

2) The chosen 45 RBDs may not fully represent all ACE2-using and non-ACE2-using sarbecoviruses available. For example, Rc-o319 from Japan, BB9904 from Bulgaria, RshSTT182 from Cambodia, RacCS203 from Thailand were not examined for ACE2 orthologs binding. For Rc-o319 it also showed bat ACE2 binding, but not *R. affinis*. In this case, the completeness of the investigation is impaired and when it comes to the ancestral sequence reconstruction, these sequences should be included and a comparison should be made to see if any conclusions would change after such sequences are added.

This is a very good suggestion—it is important to test whether our proposed evolutionary history of ACE2 binding remains robust in light of newly reported sarbecovirus sequences. In addition to the sequences listed here, several new sarbecoviruses from Europe (RhGB01, Khosta-1, Khosta-2) have recently been reported, in addition to the recently reported sequences of RsYN04, PRD-0038, and PDF-2370 (which we characterized in Fig. 4b of the original submission, but we did not test their influence on our ancestral sequence reconstruction).

To further support our key conclusion that ACE2 binding is an ancestral trait that is present in non-Asian lineages, we added substantial new data in Figure 4 and Extended Data Fig. 10. First, we

incorporated newly reported (or for BB9904, previously missing) sarbecoviruses into our phylogeny in Fig. 4a and Extended Data Fig. 10a-d (Africa/Europe sarbecoviruses: BB9904, RhGB01, Khosta-1, Khosta-2, PRD-0038, PDF-2370; uniquely branching sequence RsYN04; SARS-CoV-2-related RshSTT182, Rc-o319; Clade 2 RBD RacCS203). We inferred an updated sequence for AncSarbecovirus on this updated phylogeny, and confirmed that, like our original sequence, the updated AncSarbecovirus RBD binds to *R. affinis* ACE2 and is capable of evolving human ACE2 binding via single amino-acid mutations (Extended Data Fig. 10e).

Second, because of their important phylogenetic positions, we cloned the newly described European sarbecovirus RBDs (RhGB01, Khosta-1, and Khosta-2) into our yeast display platform and tested their binding to a panel of ACE2 orthologs. Strikingly, we found that the wildtype Khosta-2 RBD from Russia can bind to human ACE2 (Fig. 4b), the first time human ACE2 binding has been detected in a wildtype RBD from outside of Asia. Furthermore, the Khosta-1 RBD can bind to *R. affinis* ACE2 and this binding is improved by our candidate T498W mutation identified from BtKY72 (Fig. 4c). We did not detect ACE2 binding for the RhGB01 RBD, but as with BM48-31, this may be a limit of our use of SE Asian bat ACE2s when studying these viruses that circulate within different species of *Rhinolophus* bats. Taken together, our experiments on these new European sarbecoviruses support and extend our conclusions previously presented for the three African sarbecoviruses (Fig. 4c) and highlight the existence of naturally circulating sarbecoviruses in Europe with latent human ACE2 binding capability, further emphasizing the importance of our discovery of ACE2 binding and evolvability in non-Asian sarbecoviruses.

Because new sarbecoviruses are continually being reported, it will always remain a work-in-progress to maintain complete understanding of their ACE2 binding properties. As the reviewer notes, Rc-o319 is already known to utilize ACE2 (from its matched host species *R. cornutus*), and a more extensive study indicating ACE2 utilization of RshSTT182 is a subject of ongoing work in our groups. RacCS203, despite sequence similarity to SARS-CoV-2 in some genetic regions, has an unambiguously Clade 2 RBD acquired via recombination and is thus not expected to bind ACE2. Our central conclusion that ACE2 binding is an ancestral and evolvable trait has remained robust to discovery of new sequences in key phylogenetic positions (Extended Data Fig. 10e), such as RsYN04 and newly described European and African sarbecoviruses. We note that even since we began this update of our survey, more new sequences have been described, such as the BANAL sequences from Laos. However, these RBDs are very closely related to SARS-CoV-2 (only a few residue mutations) and therefore without impact on our AncSarbecovirus inference. Furthermore, more sarbecoviruses with Clade 2 RBDs continue to be described, but we emphasize that for our inference of an ancestral ACE2 binding function, it is not the sheer number of non-ACE2-utilizing Clade 2 RBDs that are sampled but rather their phylogenetic placement that matters—and our results support Clade 2 RBDs as a derived lineage with the loss of binding to all ACE2s tested thus far being a trait that evolved uniquely in the common ancestor of Clade 2 RBDs. Unless new sarbecovirus lineages are discovered that (1) lack any ACE2 binding, and (2) diverge *prior* to the divergence of Africa/Europe and Asian lineages (that is, branch off prior to our node currently labeled as the ancestral sarbecovirus), our conclusions of ancestral sarbecovirus binding to bat ACE2s and ready acquisition of human ACE2 binding are expected to remain true.

3) For Figure 3, Extended Figure 6 and related results for ACE2 binding evolution, it is better to include the RsSHC014 clade. This clade is attract great interest in vaccine development as SARS-CoV-2 antibodies generated from the current vaccines were much less effective to neutralize this clade than the other sarbecoviruses. Furthermore, this RsSHC014 clade alone can bind both *R. affinis* and *R. sinicus* from modestly to strongly while other human and civet SARS-CoV-1, “Clade 1a” and SARS-CoV-2 “Clade 1b” did not show such a pattern. Would these suggest that the RsSHC014 clade was evolutionary different from SARS-CoV-1 ad SARS-CoV-2?

We agree with the reviewer that RsSHC014-like RBDs show differences in ACE2 binding patterns compared to other SARS-CoV-1-related bat sarbecoviruses. We chose backgrounds for mutagenesis prior to obtaining our complete results across the sarbecovirus RBD panel, because our experimental platform combines all RBD and mutant sequences into a single pool for the high-throughput titration assays. Therefore, at the time of library assembly, we were not yet aware of the interesting properties we ended up seeing in RsSHC014. Because all titrations between Figures 1, 2 and 3 are performed in a single bulk experiment, additional mutagenesis and ACE2 titration assays of an additional background in the context of the current work would require re-doing all experiments. Although we could add secondary binding assays for select backgrounds, as shown in Fig. 4, these ‘isogenic’ binding assays do not scale to large numbers of mutants. That being said, (1) we have added additional discussion of the RsSHC014 results in our initial description of our ACE2 binding results, highlighting its unique pattern of ACE2 binding and suggesting its relevance for future study; (2) the primary focus was to understand the evolvability of new ACE2 binding specificities, and from that perspective, we already see RsSHC014 binds strongly to diverse ACE2s and so its future evolvability with respect to the current ACE2 panel is of less relevance; and (3) our current survey of mutations at six key ACE2-contact positions has limited utility with respect to these broader questions about antigenicity and evolutionary divergence in sequence landscapes that the reviewer proposes, but toward these goals we are currently pursuing full deep mutational scans across different sarbecovirus RBDs in which we measure the effects of all amino acid mutations at all RBD positions, and we agree RsSHC014 appears to be a very interesting background to include in these future studies.

Minor points:

For ACE2 orthologs affinity, please clarify “no”, “weak”, “modest”, and “strong” as the criteria is vague between different statements.

Thank you for this suggestion. We implemented the requested change, ensuring that all uses of such adjectives are used in relative terms within specific comparisons.

Line 53-55: How were the two distinct *R. affinis* and *R. sinicus* ACE2 alleles chosen? For each species, were the chosen two have the largest distance in the alignment of RBD-interaction residues? Please clarify.

For *R. affinis*, we used the two known RBD-interface alleles that were described in a sample of 23 *R. affinis* bats in Yunnan and Hubei, China. For *R. sinicus*, we picked two of the eight RBD-interface alleles that have been described, including an “allele 1” representative (3364) because this allele is

closest to the consensus among *R. sinicus* ACE2s, and an “allele 6” representative (1434) because this allele has been shown to hinder entry by some SARS-CoV-1-related sarbecoviruses. We have added this rationale to this section of the main text.

Line 67-68: “Binding to civet ACE2 was only detected within the SARS-CoV-1 clade whereas pangolin ACE2 binding is strongest within the SARS-CoV-2 clade”. For pangolin ACE2, Sino1-11 in the SARS-CoV-1 clade also have very strong binding, if not the strongest?

This is correct. We have changed this sentence to describe pangolin ACE2 binding as more “widespread” within the SARS-CoV-2 clade (present in three of four RBDs), rather than “strongest”.

Line 214 and Figure 4: Does the inclusion of new sarbecoviruses such as PDF-2370 and RsYN04 affect the ancestral sequence showed and discussed on Figure 3 and Extended Figure 5?

We interpret this comment as an elaboration on the reviewer’s comment (2), which asks how newly reported sarbecovirus sequences impact the sequences and phenotypes of our reconstructed ancestral sequence, to now understand whether this updated ancestor exhibits the same degree of evolvability (as Fig. 3 and the previous Extended Data Fig. 5 are about mutagenesis in the ancestral background). Full mutagenesis and repeating high-throughput titration assays with the updated ancestors is prohibitive as described above, but we did validate the key conclusion in Fig. 3, that single mutations in the reconstructed AncSarbecovirus RBD can confer human ACE2 binding. Specifically, as shown in Extended Data Fig. 10e, we confirmed that the K493Y and T498W mutations enable human ACE2 binding in the updated AncSarbecovirus sequence. Our conclusions therefore remain robust in light of newly reported sarbecovirus sequences.

Referee #2:

Tyler et al performed a comprehensive analysis of the ACE2 binding of the know sarbecovirus RBDs and their mutations by using the yeast-surface display platform, deep mutational scanning and high-throughput measurement of ACE2. The authors also reconstructed the ACE2 binding ancestor of all sarbecovirus RBDs based on one of two tested *R. affinis* ACE2 allele. The major conclusions include: 1) the human ACE2 binding is restricted to RBDs with SARS-CoV-2 and SARS-CoV-1 clade 1 and show different affinity among the tested RBDs. 2) Binding to civet ACE2 was only detected within the SARS-CoV-1 clade 1 whereas pangolin ACE2 binding is strongest within the SARS-CoV-2 clade. 3) Mouse ACE2 has a low or no binding affinity to the tested RBDs. 4) The two *R. sinicus* bat ACE2 alleles tested only interact with SARS-CoV-1 clade 1 isolates. No binding to these *R. sinicus* ACE2s was detected in the SARS-CoV-2 clade. Many SARS-CoV-2 and SARS-CoV-1 clade 1 RBDs bind to rhinolophus affinis ACE2. 5) For the first time authors detected the binding to a rhinolophus affinis ACE2 allele (discovered in China) by the RBD of a Kenya sarbecovirus BtKY72. 6) No ACE2 binding was detected for the RBDs of SARS-CoV-1 clade 2. 7) The ACE2 binding is an ancestor trait of the sarbecovirus and some of them (SARS-CoV-1 clade 2)

lost the ACE2 binding during the evolution. 7) Most of single amino-acid mutations at the six key ACE2-contact positions improved sarbecovirus RBD binding to ACE2 orthologs except the European sarbecovirus and SARS-CoV-1 clade 2. Several single aa mutations enable the BtKY72 RBD to bind to human ACE2. These results suggest that the ACE2 binding of sarbecovirus RBDs is evaluable and show high plasticity.

Overall, this manuscript presents an up-to-date overview of the binding affinity between sarbecovirus RBD and ACE2 orthologs. It's important for the spillover risk assessment of the novel sarbecoviruses in term of ACE2 receptor utilization. Although most of the conclusions are not novel to the specific research field, it's highly interesting to much broader readership considering that there are two sarbecoviruses which have caused emerging infectious diseases in the last 20 years.

Comments:

1. As the ACE2 utilization is an ancestral trait of sarbecovirus, it will be interesting to know if this feature is correlated with their evolutionary history. This reviewer suggest authors perform a molecule clock analysis using the RNA-dependent RNA polymerase all known sarbecovirus sequences and compare the ancestor reconstruction tree by the ACE2 binding.

The reviewer asks for molecular clock dating of our reconstructed ancestors. This is a difficult problem for several reasons: (1) molecular clock cannot be accurately used for a rapidly and heterogeneously evolving domain like the RBD, and therefore must be constructed on a domain like the RdRp as the reviewer suggests; but (2) the RdRp tree has a different topology than the RBD tree because of recombination, so the same set of "ancestors" are not present on each tree; and (3) molecular clock estimates are notoriously difficult to accurately calibrate for deep viral lineages, in particular due to saturation of mutations over long timescales in rapidly evolving viruses. For example, a selection of recent papers publishing molecular clock analyses of sarbecovirus evolution give estimates of the ancestral sarbecovirus ranging from >20,000 years before present (PMID 34478645) to 800 years before present (PMID 33711012). Given these caveats, we do not feel comfortable ascribing molecular clock estimates to the deep ancestral events we study in this paper without substantial method development that is beyond the scope of this work.

2. During the preparation of this manuscript, several publications have reported the ACE2 binding by the novel sarbecovirus RBDs. These novel strains have similar binding features as the Kenyan strain and bind to bat ACE2 not human ACE2. Authors please add these sequences in the alignment or phylogenetic trees if their binding analyses of the wildtype and mutations give too much work.

As described in our response to reviewer 1, comment 3, we have now added newly described RBDs into an updated phylogeny (Fig. 4a), and tested binding by newly reported European sarbecoviruses (Khosta-1, Khosta-2, and RhGB01, Fig. 4b) alongside our prior results on newly described African sarbecoviruses and the new lineage in Asia represented by RsYN04. We also incorporated newly described sarbecoviruses into an updated ancestral reconstruction of AncSarbecovirus (Extended Data Fig. 10e), which we found robustly maintains bat ACE2 binding and evolvability of human ACE2 binding. Another newly described sarbecovirus that we did not analyze is RaTG15, which is only a

single amino acid difference in the RBD from RsYN04 (S499G). We used RsYN04 because its sequence is published and publicly available, while the RaTG15 full spike sequence, at present, is only available on a non-standard gene sequence database that requires a specialized login account, and we simply obtained the RBD amino acid sequence from an alignment presented in a figure in the preprint.

3. For the BLI assay in this manuscript, this reviewer would suggest use a serially concentration of ACE2 protein instead of one concentration three times, it will be more accurate to know if the binding affinity is stable under different concentrations.

Thank you for this suggestion. We added additional BLI experiments in Extended Data Fig. 3a,b illustrating that the avidities illustrated for BtKY72 RBD binding to ACE2-Fc proteins are stable across different ACE2-Fc concentrations. We had initially set out to determine bona fide affinities for the various interactions described in this manuscript. However, the weak nature of these interactions (μM range affinities) prevented us from determining affinities due to the very high ACE2 concentrations needed (10-100 μM), which lead to non-specific binding. Instead, we opted to compare avidities of binding of dimeric ACE2-Fc constructs to RBDs immobilized at the same density. Binding responses are therefore only dependent on the strength of binding interactions and allow direct comparisons without requiring concentration series. The new results in Extended Data Fig. 3a,b illustrate that our avidity measurements are robust across replicates performed with different ACE2-Fc concentrations.

4. Phylogenetic analysis using the RdRp sequence classifies the SARS-CoV-1 lineage into two clades, clade 1 utilize ACE2, clade 2 not. Authors please unify these terms throughout the text.

Because different genomic segments have fundamentally different evolutionary histories due to recombination, there is no unifying nomenclature for clades, but rather nomenclature that applies to different non-recombining segments. The RBD clade nomenclature we use here is well described in the literature, going back to a 2017 paper from Zhengli Shi's group (PMID 29190287) for the description of "Clade 2" RBDs with large deletions and "Clade 1" SARS-CoV-1-like RBDs, which are described specifically with regards to phylogenetics/alignments of spike/RBD sequences. The Clade 1a/1b nomenclature for the newly emergent SARS-CoV-2 lineage was then elaborated by Letko et al. (PMID 32094589) who again were focused on the phylogenetic history of the RBD genomic region. We have consistently used this RBD phylogeny nomenclature (e.g., PMID: 34261126) which is emergent in the field. To avoid confusion with nomenclatures applied to phylogenetic structure of other genomic segments (e.g. RdRp which are often referred to as "Lineages"), we have ensured that we append "RBD" to any mention of a clade (e.g. "RBD Clade 2"). We have also added mention in the Introduction and Discussion to more explicitly point out the role of recombination in sarbecovirus evolution.

Referee #3:

I am pleased to review this interesting and potentially important paper about using a high throughput technique involving expression of the viral spike proteins in yeast and binding with ACE2s from different species (human, some bat species and other mammals). Your experiments to measure the binding of spike proteins from a set of different sarbecoviruses to the ACE2 receptor have resulted in some highly useful data, and as you have used this and similar techniques before it is well described in the methods (but please see the likely comments from other reviewers on the detailed experimental methods, I am not specialist in that area).

What sets this work apart from other work on receptor binding in sarbecoviruses is the generation and measurement of spike sequences which are only inferred to exist (rather than examples of what has actually been isolated in nature). The methods of the phylogenetic reconstruction are as I would expect (using more or less the software and models that I would expect) - firstly the tree is built, then a good evolutionary model is used for the ancestral state reconstruction; and alternative amino acids were also evaluated where there was uncertainty in the reconstructed ancestral sequences which is good.

Generally the paper is well written, and there is a lot of complex data to present. So it is much appreciated that you have made your code and results available on GitHub. This is a substantial body of code and results, and shows your commitment to open science. I notice that you stop short of trying to establish a binding prediction model for this paper, but you are already presenting a lot of work. I hope that the results (e.g. wt_variant_scores.csv and mut_variant_scores.csv etc) will be used by you and others in further analysis, e.g. for future viral fitness prediction models.

Minor comments

Phylogenetic methods lines 281-295: on reading this it seems that you have made the trees from only the nucleotides in the RBD region (about 600 nucleotides) ? Whilst this is enough to distinguish the main clades, it is probably contributing to the low bootstrap support on certain parts of the tree. Are the other regions of spike too divergent (or too many indels) to align well / or you are worried about recombination ? I see that you've tried it a few different ways, and also tried to assess recombination with GARD. Perhaps you can add little more clarification in this section ? (For SARS-CoV-2 and your earlier work the RBD region is approximately spike amino acids 330-350 ?, maybe just include some position numbers to give the general idea ?) But noted that you also say about the reconstructions, alternative reconstructions and robustness of phenotype in the main text.

The reviewer is correct that recombination is the reason we used the RBD alignment without being able to leverage more conserved domains for phylogenetic signal. Some portions of spike, particularly the S2 domain, are more conserved than the RBD. However, because of recombination, the S2 region cannot be used to better resolve the topology of the RBD phylogeny. This has been seen in prior

publications, and can also be seen in our phylogenies based on the entire spike protein in Extended Figure 1 and 10c,d. However, we do believe the more conserved S2 domain of spike does lend crucial support to our root placement as the divergence of Asian and non-Asian lineages (Extended Data Figs. 1 and 10a-d). We have added a sentence to the Methods discussing the reason we used an RBD alignment for our primary phylogeny. We also clarified the domain boundaries of our RBD alignment in the Methods and in the Extended Data Fig. 1 legend per the reviewer's request.

Referee #4:

This submission expands on the authors' previous Cell paper of 2020, "Deep mutational scanning of SARS-CoV-2...". In this submission, the focus is again on sarbecovirus receptor binding domains (RBDs) and their interaction with human and orthologous ACE2 receptors. Using high throughput RBD-ACE2 binding assays, the authors made several findings that are consistent with previous studies, and they also made some important original discoveries. A valuable new finding is that geographically isolated sarbecoviruses from Africa will use bat ACE2 for entry. Related to this finding are data demonstrating that sarbecovirus RBDs are "evolutionarily plastic", in that single aa substitutions markedly change orthologous ACE2 binding affinities. Together these findings point out that zoonotic sarbecovirus spillovers into humans might take place on continents beyond Asia.

The submission also highlights the utility of the high throughput RBD-ACE2 binding assays. The authors point out that complex RBD context makes sequence-based predictions of hACE2 binding nearly impossible – hence there are needs for the approaches illustrated in the submission. These approaches were shown once again to yield important information – a good example is with the analyses of RBD bindings to human and mouse ACE2, and preliminary identification of specifically mouse ACE2-binding RBDs for eventual engineering into "safe" mouse adapted viruses.

The study is very thorough and treats uncertainties carefully. There is support for a central argument that sarbecovirus utilization of ACE2 is an ancestral trait, from which ACE2 usage can be lost through deletion of RBD residues. Reconstruction of putative ancestral RBDs was considered with attention to inherent ambiguities and the results do support divergent, not convergent evolution of RBDs in Asia and Africa.

There are only minor suggestions for improvements:

(1) there could be some mention of RBD evolution in the embeco and merbeco groups, so that readers appreciate that RBD plasticity extends beyond sarbecos and ACE2 receptors

We added a description in the Discussion as to how the dynamics of RBD evolution we describe for sarbecoviruses are likely relevant to other coronaviruses that likewise show porous species boundaries.

(2) there could be some consideration of whether the clade 2 RBD deletions eliminate this domain's cell binding activity, with the NTDs then serving as principal RBDs (some literature may support this suggestion)

Although we don't know of any literature indicating whether sarbecoviruses with Clade 2 RBDs instead use S1^A/NTD attachment to enable cell entry, this is certainly a plausible hypothesis. We have added description of this possibility to this paragraph of the main text associated with the new Extended Data Fig. 4.

(3) there could be some mention of possibly frequent sarbecovirus recombinations, which could generate ACE2 "gain" in clade 2 viruses (loss and gain of "ancestral" ACE2 binding might be ongoing and frequent through the process of discontinuous viral RNA replication).

Yes, recombination is certainly a further elaboration of how ACE2 binding evolves within viral lineages. We have added mention of sarbecovirus recombination in the Introduction and the Discussion.

Reviewer Reports on the First Revision:

Referee #1

The authors have addressed all of my comments in a satisfactory manner.

Referee #2

I've no more comments.

Referee #3

Thank you for adding some more detail on the RBD sequences used in your phylogenies, and I also see that you have now updated the phylogenies and included more experimental results.

The revisions you have made in response to my previous comments seem suitable, and also the revisions made in response to the other reviewers comments are good too and improve the robustness of the manuscript.

Also, I agree with you on the response to another reviewer about not doing molecular clock analyses on diverse clades of the RBD, and the difficulty of doing it on the RdRp instead due to recombination (and still also deep time). Not having a time-scaled tree does make figure 2a in particular look slightly odd however, and perhaps rotate the branches (would have to be for figure

1 and 2 to match) - but that might not exactly work either, so it is probably not necessary to alter this.

Author Rebuttals to First Revision:

We have added a paragraph in the Methods, “Biosafety considerations,” where we lay out the biosafety concerns of our work and weigh them against the public health benefits.

No other changes were suggested in the second round of referee comments.